



# Machine learning for improvement of upper tropospheric relative humidity in ERA5 weather model data

Ziming Wang[1,2], Luca Bugliaro[1], Klaus Gierens[1], Michaela I. Hegglin[3,4], Susanne Rohs[5], Andreas Petzold[5], Stefan Kaufmann[1], Christiane Voigt[1,2]

[1]Institute of Atmospheric Physics, Deutsches Zentrum für Luft- und Raumfahrt (DLR), Oberpfaffenhofen, 82234, Germany
[2]Institute of Atmospheric Physics, Johannes Gutenberg University Mainz, Mainz, 55128, Germany
[3]Institute of Energy and Climate Research, IEK-7: Stratosphere, Forschungszentrum Jülich, Jülich, 52428, Germany
[4]Department of Meteorology, University of Reading, Reading, RG6 6ET, United Kingdom
[5]Institute of Energy and Climate Research, IEK-8: Troposphere, Forschungszentrum Jülich, Jülich, 52428, Germany

*Correspondence to*: Ziming Wang (Ziming.Wang@dlr.de)

**Abstract.** Knowledge of humidity in the upper troposphere and lower stratosphere (UTLS) is of special interest due to its importance for cirrus cloud formation and its climate impact. However, the UTLS water vapor distribution in current weather models is subject to large uncertainties. Here, we develop a dynamic-based humidity correction method using artificial neural network (ANN) to improve the relative humidity over ice (RHi) in ECMWF numerical weather predictions. The model is trained with time-dependent thermodynamic and dynamical variables from ECMWF ERA5 and humidity measurements from the In-service Aircraft for a Global Observing System (IAGOS). Previous and current atmospheric variables within ±2 ERA5 pressure layers around the IAGOS flight altitude are used for ANN training. RHi, temperature and geopotential exhibit the highest impact on ANN results, while other dynamical variables are of minor importance. The ANN shows excellent performance and the predicted RHi in the UT has a mean absolute error MAE of 6.6% and a coefficient of determination $R^2$ of 0.93, which is significantly improved compared to ERA5 RHi (MAE of 15.7%; $R^2$ of 0.66). The ANN model also improves the prediction skill for all sky UT/LS and cloudy UTLS and removes the artificial peak at RHi = 100%. The contrail predictions are in better agreement with MSG observations of ice optical thickness than the results without humidity correction for a contrail cirrus scene over the Atlantic. The ANN method can be applied to other weather models to improve humidity predictions and to support aviation and climate research applications.

## 1 Introduction

The atmospheric region of the upper troposphere and lower stratosphere (UTLS) in the tropics (Dessler and Sherwood, 2009) and the extratropics (Gettelman et al., 2011) plays a crucial role in the climate system. Within the UTLS, atmospheric humidity significantly influences the radiation budget at the top-of-atmosphere (TOA) (Riese et al., 2012). In fact, water vapor is the dominant atmospheric long-wave absorber in the context of the global greenhouse effect (Schmidt et al. 2010). Further, observed increases in stratospheric water vapor (Hegglin et al., 2014) contribute to both stratospheric cooling and tropospheric warming (Forster and Shine, 2002), and act as positive feedback to surface temperature (Tao et al., 2023). Relative humidity



over ice (RHi) > 100% or ice supersaturation is of major importance for the formation and persistence of natural cirrus and aircraft-induced contrail cirrus (Kärcher, 2018). Cirrus in this region can survive for hours if ambient conditions are ice supersaturated (Zhao et al., 2023) and they can have a positive cloud radiative effect on climate (Gazparini et al., 2020). Hence,

accurate observations and representations of UTLS water vapor (Hegglin et al., 2014) are essential for climate and weather research.

During the past decades, many humidity measurements from aircraft in situ (Krämer et al., 2009; Diao et al., 2015; Kaufmann et al., 2018), lidar (Groß et al., 2014; Krüger et al., 2022), balloon-borne (Heymsfield et al., 1998; Dickson et al., 2010; Rollins et al., 2014) and polar orbiting satellite instruments (Lamquin et al., 2012; Hegglin et al., 2013) show that high RHi may often

occur in the UT. However, these observations are limited in space and time, and the uncertainties are relatively high (Gierens et al., 2020). Important in situ humidity data can also be provided by in-service passenger aircraft (Petzold et al., 2020; Reutter et al., 2020), but three-dimensional fields of RHi and dynamics for large geographic regions are currently only available from numerical weather prediction (NWP) models, for instance the Integrated Forecasting System (IFS) at the European Centre for Medium-Range Weather Forecasts (ECMWF, 2016) and ICOsahedral Non-hydrostatic (ICON, Zängl et al., 2015; Seifert and

Siewert, 2024) at the German Weather Service. A wet bias in the RHi of the extratropical LS has been identified in the operational ECMWF IFS forecast and analysis data, as observed in comparison with in situ measurements from research aircraft (Kaufmann et al., 2018) and from Civil Aircraft for the Regular Investigation of the atmosphere Based on an Instrument Container (CARIBIC) passenger aircraft flights (Dyroff et al., 2015). In contrast, a dry bias of RHi is observed in the cloudy UT when compared with aircraft measurements in the In-Service Aircraft for Global Observing System (IAGOS) (Teoh et al.,

2022). The utilization of the saturation adjustment process (Tompkins et al., 2007), wherein supersaturation relaxes to saturation upon cloud formation, results in a systematic underestimation of both the frequency and magnitude of ice supersaturation at cruise altitudes within NWPs and global climate models (Sperber and Gierens, 2023).

There is ongoing debate on the physical explanations of the NWP humidity bias in the UTLS, which is a crucial factor to consider for the improvement of atmospheric humidity prediction. According to Kunz et al. (2014), the influences of dynamical

transport processes are challenging for the simulations of ULTS humidity distribution. Backward trajectory analyses reveal a positive relationship between the moist bias at the aircraft flight level and air masses originating from high northern latitudes in the LS (Dyroff et al., 2015). The relationships between uncertainty in atmospheric mixing and the simulated composition of water vapor in the LS, as well as the radiative consequences in the UTLS, are highlighted by Krüger et al. (2022) and Riese et al. (2012), respectively. Small-scale stratospheric intrusions, which are frequently observed in the UTLS but are unsolved

by the NWP model, are another possible source of moisture bias (Dyroff et al., 2015). Numerical diffusion, which can easily smoothen the gradients of humidity across the hydropause, is also a possible reason for the moist bias in the LS (Stenke et al., 2008). Interestingly, Woiwode et al. (2020) show that there is little dependency of the moist bias on temporal or vertical model resolution in the ECMWF IFS analysis and forecast data.

The assimilation of observations into NWP models is the state-of-the-art way to improve weather forecast (Lawrence et al.,

2019; van der Linden et al., 2020). A great deal of effort has also been focused on post-processing of NWP data to improve



the accuracy of atmospheric humidity and ice supersaturation prediction, as well as contrail cirrus, utilizing long-term aircraft measurements from IAGOS (Teoh et al., 2020; Wolf et al., 2023). Teoh et al. (2022) employ in situ measurements from IAGOS to formulate a correction method for ERA5 RHi fields. With this method, the probability density function (PDF) of ERA5-corrected RHi inside ice supersaturation closely aligns with IAGOS measurements. Another humidity bias correction, also

aiming to achieve consistency between IAGOS and ERA5 through a multivariate quantile approach, results in a notable reduction of the RHi bias (Wolf et al., 2023).

Gierens and Brinkop (2012) investigate the distributions of the dynamical quantities - divergence, relative vorticity, and vertical velocity from ECMWF IFS within and outside ice supersaturated regions and notice distinct patterns. Gierens et al. (2020) postulate that a more accurate prediction of ice supersaturation in NWP models may be achievable by further

incorporating dynamical atmospheric fields with ERA5 RHi in a general regression method. Wilhelm et al. (2022) also suggest a possibility to base an improved forecast of persistent contrails not only on the traditional quantities of temperature and RHi, but also on these dynamical proxies as well. In a recent study, Hofer et al. (2024) show that dynamical proxies taken only at the time and location of the forecast are insufficient for an improved prediction of ice supersaturation. However, they note the potential for improving RHi predictions by incorporating additional forecast data from earlier time points and upstream areas.

When improving the quality of meteorological data, machine learning techniques have been widely used nowadays. Kadow et al. (2020) have demonstrated the skill of artificial intelligence in reconstructing surface temperatures when combined with climate model data. A machine learning-based approach trained directly from historical NWP reanalysis data is introduced by Lam et al. (2023) to predict hundreds of weather variables at a remarkable speed. It outperforms the most accurate operational systems on 90% of the verification tests, even without special consideration of vertical transport. Teoh et al. (2020) also suggest

in their outlook on RHi correction that further effort can be made to explore machine learning techniques to improve the accuracy of the ERA5 humidity fields.

This paper aims at improving predictions of atmospheric humidity, in particular RHi and ice supersaturation, in the UTLS region starting from ERA5 fields using machine learning. The previous humidity corrections of Teoh et al. (2022) and Wolf et al. (2023) for ERA5 model data were based on regression fitting methods using IAGOS observations but neglected the

temporal evolution of dynamical quantities in the horizontal and vertical directions that led to the humidity bias. Targeting that gap, we develop an artificial neural network (ANN) model to correct relative humidity (and specific humidity in the supplement) from ERA5, leveraging thermodynamic conditions and dynamical quantities from ERA5, along with measured water vapor data from IAGOS above the Atlantic Ocean, Europe and Africa in 2020. The investigation is guided by three specific questions:

95        1. To what extent do atmospheric states impact the subsequent evolution of humidity fields?

       2. Is it feasible to develop a dynamic-based machine learning method to correct the humidity bias in the UTLS?

       3. How do the outcomes of the new method influence the ability to forecast ice supersaturation?

Finally, we apply the improved humidity fields for computing the optical properties of contrail cirrus in a particular situation using the Contrail Cirrus Prediction model (CoCiP) and compare the simulation results to satellite observations.



This paper is outlined as follows: Sect. 2 provides an overview of the IAGOS humidity measurements (Sect. 2.1), ERA5 and IFS data as input to the ANN model and for the application (Sect. 2.2), the collocation procedure of the measurement data with ERA5 (Sect. 2.3), and the initial comparison (Sect. 2.4), the contrail cirrus prediction model CoCiP (Sect. 2.5), and satellite remote sensing techniques for retrieving the microphysical properties of cirrus clouds (Sect. 2.6). In Sect. 3, the concept of the temporal dependence of RHi on the evolution of meteorological parameters, the development of the RHi improvement model, and the importance of the selected synoptic variables on RHi prediction are explained in detail. The evaluation of the RHi improvement model using different metrics is presented in Sect. 4. The corresponding information for specific humidity is provided in the supplement. Thereafter, Sect. 5 assesses the impact of the ANN humidity correction on the simulations of contrail cirrus in a case study. The conclusions are summarized in Sect. 6.

## 2 Data and application approaches

### 2.1 In-service Aircraft for a Global Observing System (IAGOS)

The In-service Aircraft for a Global Observing System (IAGOS; Petzold et al., 2015) is a European Research Infrastructure that implements instruments on long-range aircraft of internationally operating airlines for providing long-term in situ measurements of trace gases and meteorological conditions. These measurements are very valuable for the purpose of this study as most flight tracks are situated at heights between 9 and 13 km in the UTLS region. All aircraft within IAGOS have been equipped with a platinum sensor for temperature measurements with an accuracy of ±0.5 K, and a collocated capacitive sensor for monitoring RHi with an uncertainty of 5% to 10% (Petzold et al., 2020). The temperature detected at the sensor is transferred to the air temperature $T_{IAGOS}$ by accounting for the (incomplete) adiabatic heating and the inlet heating. RHi is then derived using the measured water vapor mixing ratio, pressure, and $T_{IAGOS}$ based on the saturation water vapor pressure equation from Sonntag (1994). The uncertainty of RHi increases with decreasing temperature due to a slower response time. In the dry conditions (RHi<10%) of the LS, the sensor has only limited accuracy (Rolf et al., 2023), so these data have been excluded for further evaluation. The temporal resolution of IAGOS measurements amounts to 4 s.



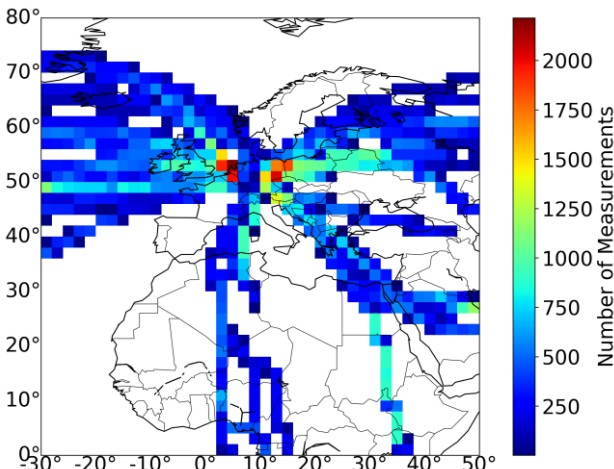

**Figure 1: Number of IAGOS measurements per 2° × 2° latitude-longitude grid box between 200 hPa and 400 hPa over the Atlantic Ocean, Europe and Africa for the year 2020. The measurements are filtered based on data quality, details see the text.**


The global distribution of IAGOS data is not uniform in every region since it is dependent on preferred flight routes and weather conditions. Europe and the North Atlantic Region (NAR) show a high density in IAGOS data therefore we focus on this domain (Fig. 1, between 30°W and 50°E). In addition, we aim to cover the latitude range between 80°N and the equator because ice supersaturation occurs very frequently in the UT of the tropics. The geographic position of the aircraft, the time, data quality flags, ambient pressure, temperature $T_{IAGOS}$ (Berkes et al., 2017), and $RHi_{IAGOS}$ from IAGOS in the year 2020 are collected to produce the output humidity data set of the ANNs. We use only the IAGOS measurements that fulfill the following criteria: IAGOS quality flag is not "limited" or "invalid", and measurements are located between 0 and 80°N, and 30°W and 50°E, and between 400 and 200 hPa.

## 2.2 ECMWF reanalysis and forecast

Meteorological data for the year 2020 in the same region in Sect. 2.1 is sourced from the ERA5 reanalysis data, obtained from the ECMWF Copernicus Climate Data Store (Hersbach et al., 2020). The assimilation system takes new observations and combines them with IFS forecast data from 12 hours before the given time to make the best estimate of the current state of the atmosphere. ERA5 data is on an equidistant latitude-longitude grid of 0.25° resolution with an hourly output on 37 pressure levels. Hourly atmospheric parameters on pressure levels between 200 and 250 hPa with a 25-hPa spacing and between 250

and 400 hPa with a 50-hPa spacing are used for model training. The IFS forecast data (137 model levels) are used for predicting contrail cirrus (see Sect. 5). Using pressure level data for ANN training reduces the size of the training dataset and saves model training time.

We use the following thermodynamic parameters from ERA5: temperature $T_{ERA5}$, and $RHi_{ERA5}$ (in the main text) or specific humidity $q_{ERA5}$ (in the supplement) as the inputs for the ANN model. The saturation water vapor pressure equation from



Alduchov and Eskridge (1996) is used here to calculate $RHi_{ERA5}$ from $q_{ERA5}$. In addition, specific cloud ice water content *ciwc* from ERA5 is utilized to differentiate between the cirrus- and cirrus-free regions. As shown in Table 1, this study also considers dynamical parameters, including geopotential (*z*) for adiabatic shifts, vertical velocity (*w*) in Pa/s representing vertical air mass motion, divergence (*d*) indicating air spread or convergence, horizontal wind speed (*u* and *v*) in m/s, and relative and potential vorticity (*vo* and *pv*) characterizing air rotation and stratosphere-troposphere exchanges. The use of *pv* specifically helps identify the dynamical tropopause and distinguish between UT and LS.

**2.3 Data collocation**

**Table 1: Overview of the variables used in this study. Spatial resolution of ERA5: 0.25°. Vertical resolution of ERA5 on pressure levels: 25-50 hPa. The original temporal resolution of ERA5 and IAGOS: 1 h and 4 s. Study regions: Atlantic, Europe and Africa.**

| source | variable (* used in the RHi ANN) | description | unit |
|---|---|---|---|
| ERA5 data<br>pressure level<br><br>• at the current time and level<br>• 2h and 6h before the current time and at the current level<br>• at the current time and l, 2 level above and below the current level | * $T_{ERA5}$ | air temperature | K |
| | *ciwc* | specific cloud ice water content | Kg/kg |
| | * $RHi_{ERA5}$ | relative humidity w.r.t. ice | % |
| | $q_{ERA5}$ | specific humidity | g/Kg |
| | * $z$ | geopotential | $m^2/s^2$ |
| | * $w$ | vertical velocity | Pa/s |
| | * $d$ | divergency of wind | $s^{-1}$ |
| | * $u$ | eastward component of wind | m/s |
| | * $v$ | northward component of wind | m/s |
| | * $vo$ | relative vorticity | $s^{-1}$ |
| | $pv$ | potential vorticity | $s^{-1}$ |
| | time | hour | 1 |
| | level | pressure | hPa |
| IAGOS<br>at the current time | * $RHi_{IAGOS}$ | relative humidity w.r.t. ice | 1 |
| | $T_{IAGOS}$ | air temperature | K |
| | pressure | air pressure | Pa |

The ERA5 grid boxes that are closest to the IAGOS observations in both time and space are selected to align with the IAGOS measured humidity and temperature data sets between 400 hPa and 200 hPa. In contrast to other studies (Wolf et al., 2023; Hofer et al., 2024), this study also uses the temporal evolution of meteorological conditions before the IAGOS observation time. Specifically, RHi in the UTLS is influenced by horizontal and vertical air motions like air mass uplift (Diao et al., 2015)



in convection or frontal systems, or stratospheric intrusions. To account for these, thermodynamic and dynamical data up to 6
h prior to the IAGOS data acquisition time, with 1h intervals, and within two pressure levels above and below the IAGOS
acquisition location, are linked with $RHi_{IAGOS}$ at the IAGOS acquisition time and location. These ERA5 variables are vertically
interpolated to match the IAGOS location based on pressure levels. While the ERA5 data retains its original temporal and
spatial resolution, the water vapor data measured by IAGOS, which includes several data points within a single ERA5 grid
box, is averaged. This averaging reduces the autocorrelation in the measured data due to the response time of the sensor,
accounts for internal ERA5 grid box variability, and maintains a proportion of ice supersaturation after averaging. This
collocation of model meteorological variables and measured humidity values from the year 2020 comprises 3.99 million
individual data points, from which 80%, 10%, and 10% are randomly selected for training, testing during the model
development, and validating the ANNs, respectively. This method copes with considerable variability and sharp gradients in
the humidity fields, and thus can help to estimate realistic atmospheric humidity distributions for comparisons and model
application.

## 2.4 Initial ERA5 RHi evaluation using IAGOS

We first compare and quantify the difference between ERA5 and in situ measurements provided by IAGOS with respect to
temperature, and specific and relative humidity. RHi in cirrus clouds in NWP can have a low bias due to the application of
saturation adjustment in cloud parameterizations (ECMWF, 2016), hence we differentiate between clear (cloudy) conditions
using *ciwc* equal to zero for all the current and ±2 pressure layers from ERA5. We further distinguish between UT (LS)
dependent on the threshold *pv* smaller than 2 PVU. This means, we consider the dynamical tropopause as done for instance in
Reutter et al. (2020). In the following, we focus on RHi (for comparison of other parameters, see also Sect. S1 and S3 in the
supplement).



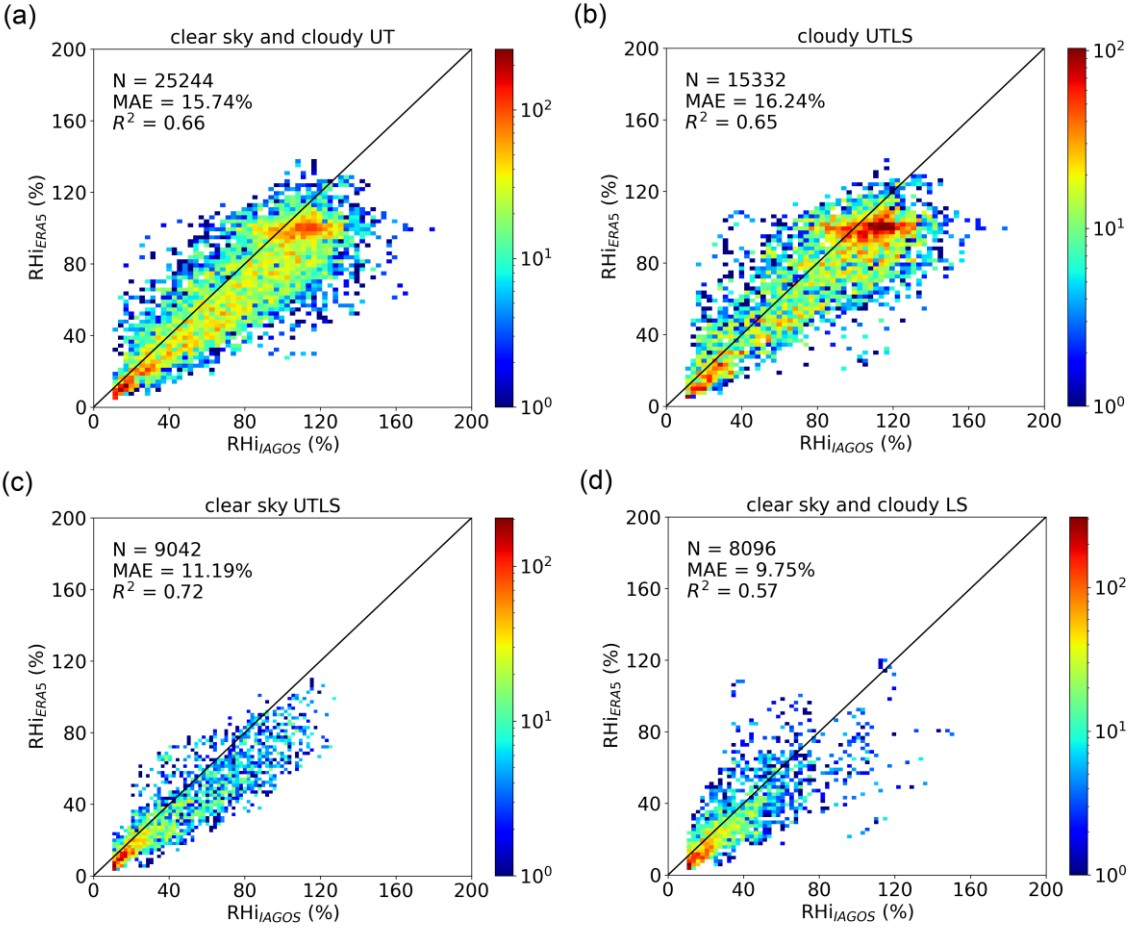

**Figure 2: Comparisons of $RHi_{ERA5}$ against $RHi_{IAGOS}$ in (a) clear sky and cloudy UT, (b) cloudy UTLS, (c) clear sky UTLS, and (d) clear sky and cloudy LS in the test data set between 200 hPa and 400 hPa over the Atlantic, Europe and Africa for the year 2020.**

Figure 2 shows the comparison of $RHi_{ERA5}$ and $RHi_{IAGOS}$ separated either between upper troposphere or lower stratospheric conditions or between cloudy or clear sky conditions, respectively. Here we use the validation dataset from the ERA5-IAGOS collection created in Sect. 2.3, which is the same dataset used for validating the ANN model in Sect. 4.1. In the all sky UT (Fig. 2a) and in the cloudy UTLS (Fig. 2b), ERA5 RHi data show a considerably dry bias compared to IAGOS data, with mean absolute errors (MAEs) of 15.74% and of 16.24%, respectively. RHi and the magnitude of ice supersaturation in ERA5 are underestimated. In addition, an artificial occurrence accumulation peak exists in the ERA5 data set at $RHi_{ERA5}$ = 100%, which is caused by the cloud saturation adjustment. This peak is not found in the IAGOS data set. Nevertheless, $RHi_{ERA5} > 100\%$ is also observed, either in partly cloudy model boxes or in clear sky boxes, where only a fraction of the box is cloudy (with $RHi_{ERA5}$=100%) and the clear sky part is supersaturated, due to the time required for the ice nucleation process. Consequently, $RHi_{ERA5}$ values greater than 100% can occur in cloudy conditions as well. In the clear sky UTLS (Fig. 2c) and in the all sky



LS (Fig. 2d) regions, MAEs are 11.19% and 9.75%, respectively. In the all sky LS (Fig. 2d), few RHi data > 100% have been measured by IAGOS, with most observations concentrated at low RHi values. In general, the extent and the degree of ice supersaturation underestimated by ERA5 are in line with the findings by Dyroff et al. (2015) for ECMWF analysis and forecast data. The comparison of ERA5 and IAGOS RHi serves as the motivation for our study, aiming to improve the humidity prediction by NWPs.

## 2.5 Contrail cirrus prediction model CoCiP

The Contrail Cirrus Prediction Model (CoCiP) is used to predict the contrail cirrus cover and examine the contrail radiative forcing induced by individual flights (Schumann, 2012; Schumann et al., 2017, 2021b; Voigt et al., 2017, 2022; Teoh et al., 2024). The contrail model uses traffic data from the North Atlantic Tracks for the Shanwick Oceanic Control Area. When the contrail formation threshold is met by two successive flight waypoints, a contrail segment forms. Contrail initial water content, width, and depth are determined by aircraft properties and emissions (non-volatile particulate matter). Plume dispersion is a function of turbulence, wind shear, and induced heating. RHi inside contrail plumes is set at saturation, and ice water content of contrails grows or decreases in response to the ambient humidity. A Runge-Kutta integration simulates the contrail evolution until its end of life by ambient drying or particle losses from aggregation and sedimentation. The contrail life ends when the maximum contrail lifetime of 24 hours is reached, the ice number concentration is less than the background ice nuclei ($< 10^3$ m$^{-3}$), or the ice optical thickness (IOT) is less than $10^{-6}$. CoCiP simulations account for humidity exchange between contrails and the background air, and the overlap of contrails above or below clouds present in the meteorological data from the NWP forecast (Schumann et al., 2021a). The CoCiP limitation in comparison to general circulation models is its absence of atmospheric interaction and feedback (Chen et al., 2012; Burkhardt et al., 2018; Bickel et al., 2020). In Sect. 5 of this study, we present an exemplary application of the model for RHi correction derived in this paper to contrail simulations and show its effect on contrail properties. This is done by performing two CoCiP runs: for the reference run, we use NWP data from ECMWF IFS for the contrail case on 14 April 2021 over the NAR within the ECLIF3 campaign (Märkl et al., 2024), for the second run we correct the NWP humidity data with the ANN proposed in this work in the same situation.

## 2.6 Satellite remote sensing

CoCiP simulations are compared to spaceborne data from the SEVIRI radiometer aboard the geostationary Meteosat Second Generation (MSG) satellite in Sect. 5. To derive ice cloud properties, CiPS (Cirrus Properties from SEVIRI, Strandgren et al. 2017) is used. It consists of a set of ANNs trained on SEVIRI thermal observations, CALIPSO (Cloud Aerosol Lidar and Infrared Pathfinder Satellite Observations) cloud products, ECMWF ERA-Interim surface temperature data and auxiliary data, to retrieve IOT for identified cirrus clouds. Specifically developed for thin cirrus clouds, CiPS has been validated against CALIPSO, achieving detection rates of 20%, 70%, and 85% for ice clouds with IOT values of 0.01, 0.1, and 0.2, respectively. For IOT between 0.35 and 1.8, CiPS demonstrates a MAE smaller than 50%, and MAE increases for IOT values between 0.07 and 0.35.



## 3 ANN model development

### 3.1 The temporal dependence of measured humidity on individual meteorological parameters

Which meteorological parameters and at which time and pressure level should be chosen for training the RHi improvement model? To answer this question, the dependence of measured RHi on meteorological variables at preceding times and surrounding pressure levels is considered by reviewing the sources of RHi bias in the UTLS within ECMWF data. As outlined in Dyroff et al. (2015), this bias is linked to air masses residing near the aircraft's flight level of approximately 230 hPa in high northern latitudes, likely influenced by airmass vertical intrusions and horizontal transport. This points out the connection between RHi and the temporal evolution of meteorological parameters.

Based on the physical definition of RHi, a negative correlation between $RHi$ and $T$ is expected because RHi is the ratio of the partial vapor pressure of water vapor to the saturation vapor pressure with respect to ice, the latter of which increases with temperature. For dynamical variables, the positive relationship between increased geopotential $z$ values and $RHi_{IAGOS}$ is in accordance with the findings of Wilhelm et al. (2022). In addition, parameters such as vertical wind $w$, divergence $d$, horizontal wind speed components $u$ and $v$, relative vorticity $vo$, and potential vorticity $pv$ help represent the dynamical conditions at a given time and place that influence relative humidity in the model. For instance, an upward motion (negative values of $w$ in ERA5) results in cooling and a decrease in RHi, and promotes ice supersaturation. A relatively strong horizontal airmass movement with large divergence is typical for ice supersaturation. Large negative values of vorticity in anticyclonic systems are again also typical for supersaturation (Gierens et al., 2020). These connections suggest the potential to improve the RHi prediction by considering not only traditional thermodynamic variables like temperature but also dynamical proxies and their temporal evolution.

To balance information richness and modeling efficiency, only the current IAGOS humidity fields, and ERA5 data at the current time of the IAGOS measurement, a 2-h and 6-h time lag prior to IAGOS data acquisition, as well as ±1, ±2 pressure layers from ERA5 are selected as input variables to account for the typical lifespans of water vapor transport mechanisms, including deep convection, warm conveyor belt uplift regimes, and slow ascending flows. Notably, $pv$ is not provided in ECMWF model level data and subsequently excluded from the input data set during further training of the ANN model. Summarising, the ANN model is trained with the variables shown in Table1. The relevance of each input atmospheric variable to the RHi prediction model developed in Sect. 3.2 is discussed in Sect. 3.3.

### 3.2 Development and training of the ANN model for humidity improvement

An ANN is composed of a large number of units that exchange information with each other, in a similar structure and function as neural networks in human brains. A basic ANN model contains three types of layers: an input layer, one or more hidden layer(s), and an output layer. Each layer is made up of neurons. Neurons receive the weighted sum of the results of the previous layer's neurons, use it as the argument of an activation function, and forward the results to the following layer. The feed-forward ANN used in this study employs a learning technique called back-propagation, where the outputs are compared to the



target values to calculate the differences in the form of loss function. The error is then fed back to modify the weights and bias of each neuron based on the optimization method (see e.g. Ma et al., 2020).

Here, the ANN model for RHi is trained using a large set of atmospheric variables obtained from ERA5 reanalysis and RHi measured from IAGOS as explained in the previous section. The ANN learns to reproduce the nonlinear statistical relationships between the selected series of meteorological variables and humidity fields iteratively adjusting its parameters until it can robustly and accurately predict RHi in the UTLS. This procedure does not only consider RHi>100% to investigate ice supersaturation but also the full range of RHi to provide sufficient data for model building.

Based on the temporal dependence of measured humidity on individual meteorological parameters discussed in Sect. 3.1, the input variables for the ANN encompass $RHi_{ERA5}$, $T_{ERA5}$, and $z$, $w$, $d$, $u$, $v$, and $vo$. They are extracted from the ERA5 fields (Sect. 2.2) at the time of the IAGOS observation, as well as 2 h and 6 h prior, at the geographical and vertical location of the IAGOS measurement, along with ±1, ±2 ERA5 pressure levels. The output (target) variable of the ANN is $RHi_{IAGOS}$. The ANN consists of 56 inputs: 8 meteorological variables multiplied by 2 time lags (-2 h and -6 h) and 5 pressure levels (-2, -1, 0, +1, +2 pressure levels with respect to ERA5 at the current time). They are summarised in Table 1. We use 2 hidden layers with 100 neurons each, and call the humidity output $RHi_{ANN}$. The Rectified Linear Unit (Relu) serves as the activation function for the hidden layers, while a linear function is used for the output layer. The mean squared error (MSE) is adopted as a loss function, and the ANN model is optimized with a learning rate of 0.001, decay of $10^{-5}$, and momentum of 0.99 after several tests. Training of the ANN model is executed with batch sizes of 1024 and 2000 epochs. From the ERA5-IAGOS collection in Sect. 2.3, 90% of the samples are randomly allocated for model training and the remaining 10% are reserved for test purposes. The trained model is validated against the test data set of ERA5 and IAGOS, which has already been used for the comparative analysis of ERA5 and IAGOS in Sect. 2.4.

Details on the preparation of training and validation data particularly for specific humidity $q$ are provided in Sect. S2 of the supplement. Similarly, an ANN for $q$ is implemented, with *RHi* replaced by $q$ everywhere in the both input and output layers (refer to the supplement for more information). The ANN model can then be applied to ERA5 data for humidity correction in the UTLS region. The computational time required for each scene in Fig. 1 is approx. 5 seconds on a standard laptop (Intel I5 8250U CPU; 8G memory). This technique incorporates thermodynamic and dynamical meteorological values to account for the vertical and horizontal transport of water vapor and its temporal evolution, and takes advantage of numerous humidity measurements.

**3.3 Importance of the individual variables for the quality of ANN RHi prediction**

The ANN model is interpreted with an investigation of the relative contributions of input variables to the predicted $RHi_{ANN}$. $K_x$ is the relative change in loss when one input, i.e. one feature of ERA5 is set to zero for the complete input data set but the rest of the input features is kept unchanged:

$$K_x = \frac{L_x - L_0}{L_0} \qquad (1)$$



where $L_x$ is the loss (MSE) for the test data set compared with IAGOS when setting one input of ERA5 feature to zero, and $L_0$
is the loss for the full test data set. Low values of $K_x$ indicate a small impact of the change in input quantity on the output
accuracy, or vice versa.

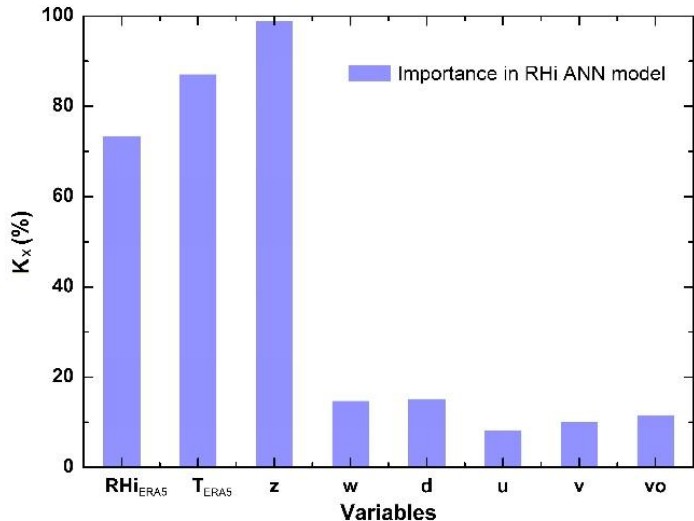

**Figure 3: Relative importance of the individual variables to the ANN model for predicting RHi.**

In Fig. 3, the importance ($K_x$) analysis for all input variables (the current time and level, i.e. -2 or -6 h, and -2, -1, +1, +2 levels
above/below) in the ANN model reveals that $RHi_{ERA5}, T_{ERA5}$, and $z$ hold the highest level of significance and carry
considerable weight among all parameters. The particular relevance of these three variables can be explained by the inherent
relationship wherein RHi typically rises in regions with decreasing temperature at higher geopotential $z$ in the troposphere.
Water vapor is carried to upper altitudes with the accompanying adiabatic cooling, which increases RHi. Thus, the ANN model
already captures the $RHi_{IAGOS}$ effectively using $RHi_{ERA5}, T_{ERA5}$ and $z$. However, although the other (dynamical) variables are
less important for the prediction of $RHi_{ANN}$, they provide a non-negligible contribution to the accuracy of the RHi prediction
model. In fact, $w$ and $d$ show weights around 15% while those for $u$, $v$, and $vo$ are 8%, 10%, and 12%. There is generally no
significant contrast in the contributions of the variables representing dynamical quantities, aligning with the findings in Hofer
et al. (2024) based on meteorological variables from the given time.
The fact that dynamical variables are important for the description of the physical processes that lead to the decrease/increase
of relative humidity but at the same time show only a moderate importance in the ANN model, could be attributed to the
correlation with other variables, the significant overlap between the conditional distributions of $RHi_{ERA5}$ on ice supersaturation
from $RHi_{IAGOS}$ or not (Hofer et al., 2024), and the ANN evaluation measure $K_x$ (Piontek et al., 2021). While the developed
ANN model comprises two hidden layers, the primary influence on $K_x$ stems from the input variables in the first hidden layer.



However, weak connections from the first hidden layer to the second might be amplified in the last step from the second hidden

        layer to the output layer.

## 4 Model evaluation and results

### 4.1 Validation of ANN RHi in clear and cloudy conditions in the UTLS

This study aims to use the ANN model to resolve biases inherent in NWP model output evaluated in Sect. 2.4. To quantify the

accuracy of the ANN model, $RHi_{ANN}$ (and $q_{ANN}$ in the supplement) is evaluated based on the test data set under four

conditions: all sky UT, cloudy UTLS, clear sky UTLS, and all sky LS. Validation results of $RHi_{ANN}$ are shown in Figs. 4, 5,

and 6 and of $q_{ANN}$ in Sect. S3, respectively.

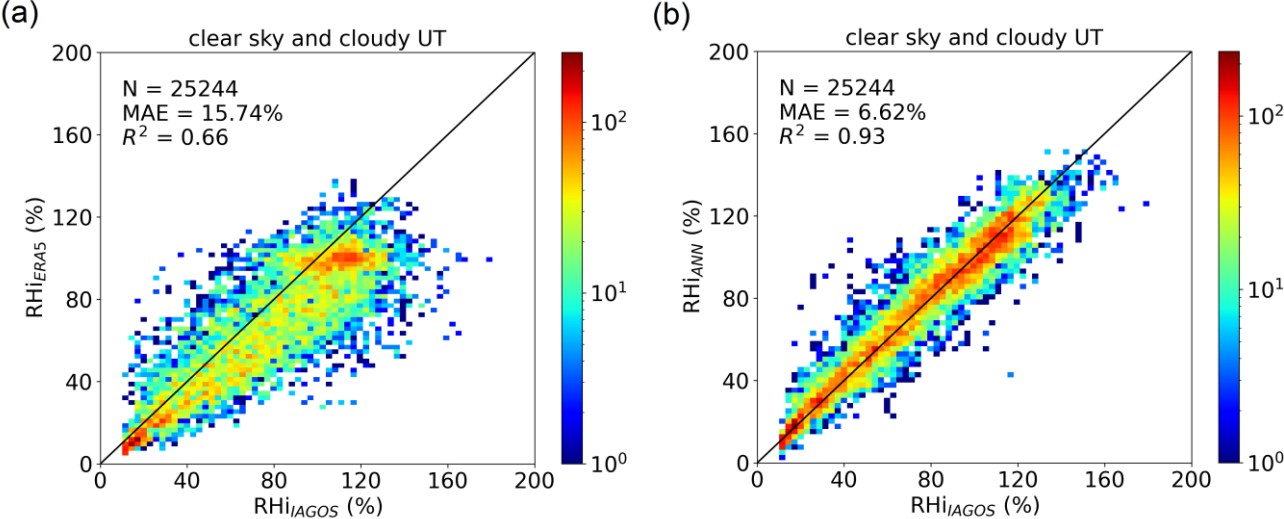

**Figure 4: Distribution of (a) $RHi_{ERA5}$ and (b) $RHi_{ANN}$ versus $RHi_{IAGOS}$ in the UT in all sky (clear and cloudy conditions) in the test**

**data set. The number of data sets N, the mean absolute error MAE and the coefficient of determination R² are shown in the panels.**

In the UT all sky condition important for cirrus clouds and contrails, a high number of measurements, comprising 25244 data

points, are used for the inter-comparison between ERA5 and the outputs of the ANN model. $RHi_{ANN}$ (Fig. 4b) demonstrates a

better agreement with $RHi_{IAGOS}$ compared to $RHi_{ERA5}$ (Fig. 4a). In particular. $RHi_{ANN}$ shows consistent values to $RHi_{IAGOS}$ at

RHi>100% which is a major improvement in comparison to the ERA5 data set. $RHi_{ANN}$ exhibits a significant higher

correlation with $RHi_{IAGOS}$ for all uncertainty parameters (mean absolute error MAE, coefficient of determination R²) compared

to ERA5. The MAE decreases significantly from 15.74% (ERA5) to 6.62% (ANN), the R² values increase from 0.66 (ERA5)

to 0.93 (ANN), and the root mean spare error RMSE decreases from 15.45% (ERA5) to 6.44% (ANN). Notably, the ANN

model also effectively corrects the existing peak at $RHi_{ERA5} = 100\%$ in Fig. 4a, and does not show a peak at RHi ~100%,

similar to the IAGOS measurements. Hence the ANN exhibits a significant improvement of RHi that would be beneficial for



cirrus and cloud predictions. For other scenarios, such as the cloudy UTLS, the clear sky UTLS and the all sky LS between 400 and 200 hPa, the comparison of RHi is shown in Fig. 5. Notably, also in the cloudy UTLS, $RHi_{ANN}$ results (Fig. 5b) exhibit a closer correlation with $RHi_{IAGOS}$ than those in the cloudy region of $RHi_{ERA5}$ (Fig. 5a). In the cloudy (Fig. 5a-b) and clear sky (Fig. 5c-d) conditions in the UTLS, the MAE of the RHi decreases from 16.24% (11.19%) to 6.77% (4.55%), respectively.

Also, the R² increases by 0.28 (0.23) to 0.93 (0.95) for the two scenarios. Again, the peak at 100% in the $RHi_{ERA5}$ distribution in the cloudy UTLS disappears in the $RHi_{ANN}$, in line with the IAGOS observations. In particular, the ANN model shows a very good performance in the cloudy UTLS, and $RHi_{ANN}$ and $RHi_{IAGOS}$ align close to the 1:1 line. While the majority of data in the cloudy UTLS is allocated at RHi>80%, some clouds were observed in ice sub-saturated conditions, or ice particles had sedimented into dryer air.

The ANN model also has strong skills of RHi correction in the LS, see Fig. 5e and f. $R^2$ values increase from 0.57 (ERA5) to 0.92 (ANN), similar to the UT region in Fig. 4. The improvement of RHi prediction by the ANN is also documented by the decrease of MAE by 5.42%. The ANN model successfully learns interconnections within the data, as evident by its more accurate $RHi_{ANN}$.

Figure 6 presents a detailed relative comparison (mean bias error MBE) of either $RHi_{ERA5}$ or $RHi_{ANN}$ as a function of

$RHi_{IAGOS}$. In Fig. 6a, the occurrences of $RHi_{ERA5} > 105\%$ are underestimated compared to the distribution of $RHi_{IAGOS}$. In Fig. 6b, the distribution of $RHi_{ANN}$ closely resembles $RHi_{IAGOS}$, showing a smoother distribution around RHi of 100%. In Fig. 6c, $RHi_{ERA5}$ shows an increasing dry bias in the UT, reaching 37% at RHi > 120%. The few data points at RHi > 120% even exhibit larger deviations by more than 60% within ERA5. As opposed to this, Fig. 6d shows that $RHi_{ANN}$ and $RHi_{IAGOS}$ have a closer agreement, with an MBE of about ±13% for all UT measurements up to 140%. The RHi between 80% and 130% in

the important range for cirrus clouds is well represented by the ANN with an MBE better than +/- 10%. This suggests that the saturated region of RHi, which presents a requisite environmental condition for new ice crystal nucleation and subsequent growth, can be more accurately parametrized.

Wolf et al. (2023) developed a humidity correction technique for $RHi_{ERA5}$ using IAGOS measurements through a multivariate quantile method. The differences between the corrected ERA5 data and $RHi_{ERA5}$ in cloudy regions are documented in their

Table 3, with the mean absolute difference ranging from -2.2% to 12.08% depending on cloud fraction. Although not directly comparable within the same timeframe, $RHi_{ANN}$ also shows a good performance, with a MAE of approximately 6.7% in the same region of all sky UT and cloudy UTLS.

$RHi_{ANN}$ also shows better agreement with independent airborne measurements compared to ERA5 data. For detailed information on the humidity data on 21 July 2021 during the CIRRUS-HL campaign, please refer to Sect. S4 in the supplement,

which includes measured data from AIMS (Atmospheric Ionization Mass Spectrometer, Kaufmann et al., 2016) instrument and $RHi_{ERA5}$ and $RHi_{ANN}$.

The investigations related to validating ANN $q$ in clear and cloudy conditions in the ULTS are shown in Sect. S3 in the supplement.






**Figure 5: Comparison of $RHi_{ERA5}$ (left column) and $RHi_{ANN}$ (right column) against $RHi_{IAGOS}$ in the (a) and (b) cloudy UTLS, (c) and (d) clear sky UTLS, and (e) and (f) clear sky and cloudy (or all sky) LS regions in the test data set. The number of data sets N, the mean absolute error MAE and the coefficient of determination R² are indicated in the individual panels.**





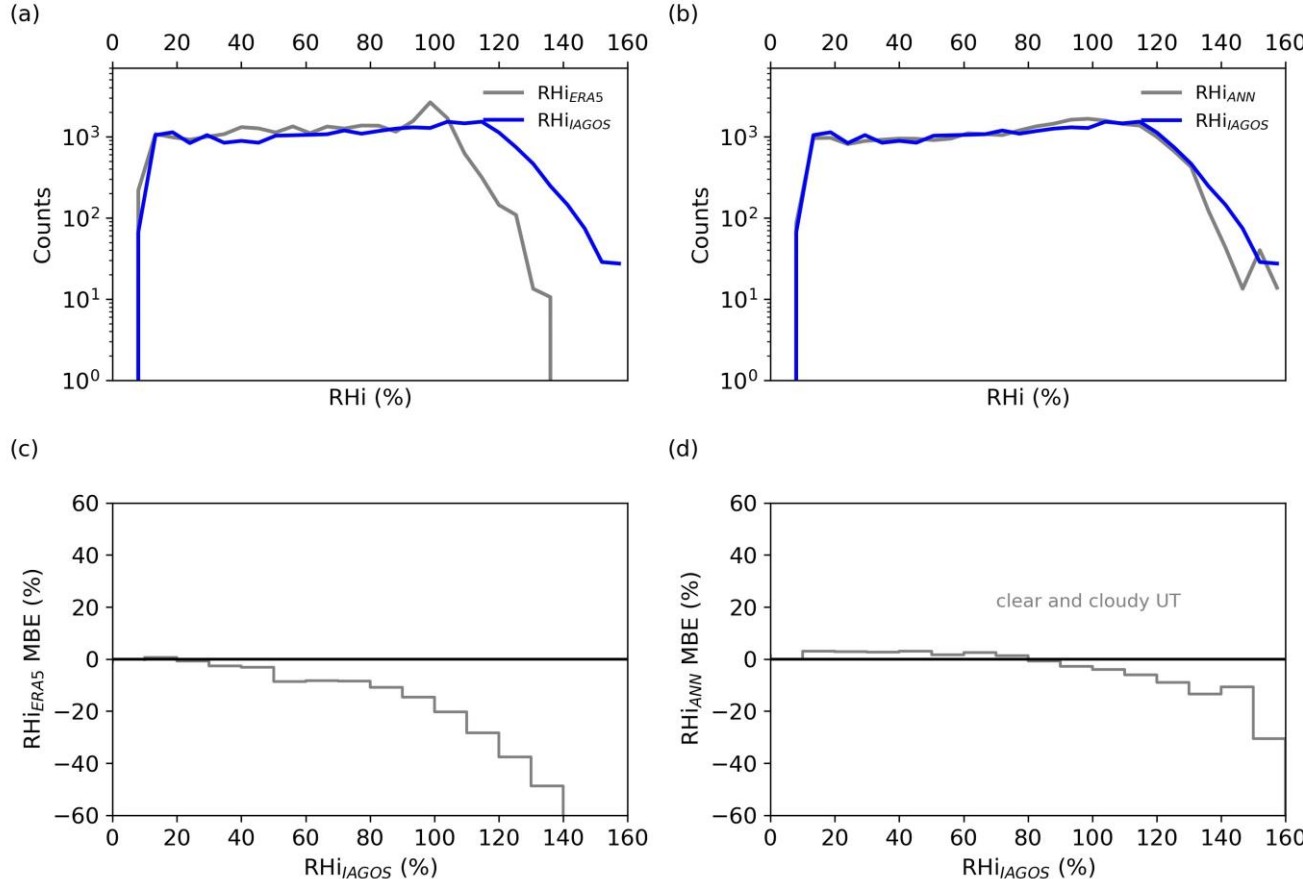

**Figure 6: Frequency distribution (a and c) and overall mean biased error MBE (%) (b and d) of $RHi_{ERA5}$ and $RHi_{ANN}$ against $RHi_{IAGOS}$ in the clear and cloudy UT (grey) in the test data set.**

## 4.2 Skill of ANN and ERA5 prediction of RHi>100% versus IAGOS data

For cirrus clouds and contrails, accurately representing RHi>100%, and thus ice supersaturation, is of great importance. Hence, we focus here on the data sets with RHi>100%. The skill of ice supersaturation prediction from $RHi_{ERA5}$ and $RHi_{ANN}$ is evaluated based on the equitable threat score (ETS), as described in Gierens et al. (2020). The ETS measures forecasting performance by assessing the proportion of correctly forecasted events and is often used in weather forecast verification (Wang, 2014). First, events are labeled according to the contingency table, with a (ice supersaturation predicted and observed), b (no ice supersaturation predicted but observed), c (ice supersaturation predicted but not observed), and d (ice supersaturation neither predicted nor observed). The ETS is then calculated using the following equations:

$$ETS = \frac{a-r}{a+b+c-r} \qquad (2)$$

with



$$r = \frac{(a+b)(a+c)}{a+b+c+d} \tag{3}$$

The ETS value gets larger when the ice supersaturation prediction is closer to the measured values (here IAGOS data). ETS = 1 indicates that all $RHi_{IAGOS}$ perfectly agree with $RHi_{ERA5}$ or $RHi_{ANN}$. ETS = 0 means a completely random distribution, while negative ETS implies a negative correlation.

Table 2: ETS values for the prediction of RHi>100% from $RHi_{ERA5}$ and $RHi_{ANN}$ in the test data set between 200 hPa and 400 hPa over the Atlantic, Europe and Africa in 2020.

| Scenarios | ERA5 | ANN |
|---|---|---|
| clear and cloudy UT | 0.22 | 0.64 |
| cloudy UTLS | 0.25 | 0.65 |
| clear and cloudy LS | 0.14 | 0.65 |

Table 2 shows ETS values for both the ERA5 and the ANN predicted ice supersaturation across all test data set in Sect. 2.4. The scores for ERA5 in all sky UT, cloudy UTLS, and all sky LS classes are 0.22, 0.25, and 0.14, respectively, indicating limited predictive skill, particularly in the all sky LS region. In contrast, the ANN model significantly enhances the ice supersaturation prediction, yielding scores of 0.64, 0.65, and 0.65 for the respective regions. This represents an approximate 0.4 increase in ETS across all classes, thereby facilitating related studies on the formation and persistence of cirrus clouds. The clear sky UTLS region is not discussed here, as we are focusing on RHi > 100%. According to Fig. 5c, few ERA5 data points fall within the ice supersaturation region.

Teoh et al. (2022) developed a statistical approach to correct the ERA5 humidity fields in particular for ice supersaturation, with the aim of adjusting the PDF of $RHi_{ERA5}$ in order to achieve a similar PDF as $RHi_{IAGOS}$. After applying this humidity correction method, the ETS for ice supersaturation in the all-sky UTLS reached a value of 0.424 when compared to IAGOS measurements in 2019, as shown in their Table S4. This statistical method outperforms $RHi_{ERA5}$ in predicting ice supersaturation, as the latter achieves an ETS of approximately 0.2. In a recent study, Hofer et al. (2024) use $RHi_{ERA5}$, $T_{ERA5}$ and dynamical proxies only from the given time in several regression models to predict RHi>100%, and find the best regression with an ETS of 0.378 for 16588 flights of the Measurement of Ozone and Water Vapour on Airbus In-service Aircraft (MOZAIC) between 2000 and 2009. While not directly comparable within the same time frame, $RHi_{ANN}$ excels in forecasting ice supersaturation relative to ERA5 and methods from Teoh et al. (2022) and Hofer et al. (2024), demonstrating a higher accuracy with an ETS as high as 0.65.

## 5 CoCiP predictions and MSG contrail cirrus observations

As an application, this section investigates the impact of improved humidity prediction in the UT on the estimation of contrail cirrus optical thickness using CoCiP simulations (Sect. 2.5) and compares the results with retrieved IOT from MSG



observations using the CiPS algorithm (Sect. 2.6). The selected case is from 10 UTC on 14 April 2021 (during the ECLIF3 campaign) over the North Atlantic Region (NAR, Sect. 2.1), representing typical contrail cirrus situation just off the coast of
Ireland. The MSG observation scene is from 09:45 UTC, as SEVIRI scans from the south, taking about 12 minutes per scene and reaching the upper edge (near the NAR) around 09:56 UTC, which is more consistent with the CoCiP simulation time. For the CoCiP simulations, specific humidity $q_{IFS}$ and other atmospheric trace gas profiles from the ECMWF IFS model level data between level 73 (about 190 hpa) and level 90 (about 400 or 410 hpa) are used as input to the ANN model to produce $q_{ANN}$ profiles. We perform two CoCiP experiments: the $q_{IFS}$ and the resulting $q_{ANN}$ profiles serve as input for CoCiP,
respectively, while other parameters, particularly cloud liquid water content and *ciwc*, are kept constant using IFS values to include the same natural cloud effects in both CoCiP simulations.

Figure 7 provides a comparison of the spatial distributions and frequency of occurrence (histogram) of CoCiP-simulated and MSG-observed IOT. In the MSG scene (Fig. 7a), contrails and contrail cirrus, represented by linear structures with IOT ~ 0.3-0.5, are situated above the Atlantic Ocean, extending from west to east and surrounded by thicker cirrus clouds with higher
IOT (even exceeding 1.0). The simulated IOTs (Figs. 7b and c) show patterns of higher IOT surrounded by lower IOT cirrus as in the MSG observations, although the single structures are not directly comparable. In addition, due to satellite detection limitations, CiPS cannot capture the thinnest ice clouds. For these reasons, a quantitative pixel-to-pixel comparison between the CoCiP simulations and the MSG observation is not meaningful in this context and we thus rather consider frequency distributions of IOT in the following. In general, the simulated IOT using $q_{ANN}$ (Fig. 7b) is closer to CiPS retrieved IOT
compared to the CoCiP simulation with $q_{IFS}$ (Fig. 7c). The histogram (Fig. 7d) shows decreasing frequencies of occurrence with increasing IOT. The better agreement is also exhibited between IOT from CiPS and the CoCiP simulation using $q_{ANN}$ than that using $q_{IFS}$. For natural cirrus, larger IOT (~0.75) is observed, while the smaller IOT (< 0.5) for contrails is of particular interest. In the lower panels showing only contrails, the simulation with increased humidity exhibits larger IOT (Fig. 7e) compared to that without humidity correction (Fig. 7f). Higher IOT up to 0.8 from CoCiP with $q_{ANN}$, compared to IOT values
below 0.5 with $q_{IFS}$, is due to the growth of contrail ice crystals from the increased amount of available water vapor in $q_{ANN}$ and is also evident in the frequency analysis (Fig. 7g).



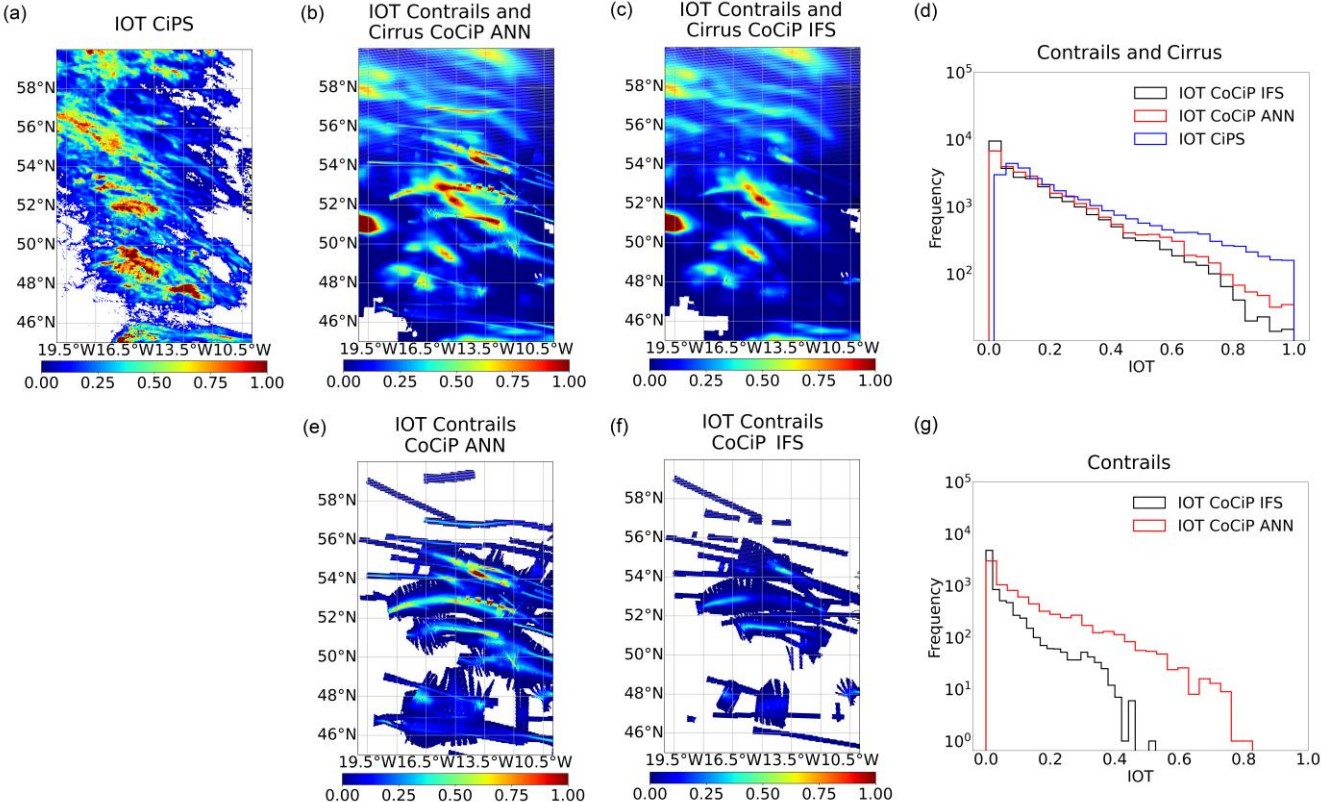

**Figure 7: Distributions of IOT for contrails and cirrus retrieved from (a) MSG observations using the CiPS algorithm, and simulated using the CoCiP model with (b) $q_{ANN}$ or (c) $q_{IFS}$ at 10:00 UTC on 14 April 2021. The IOT distribution for contrails from CoCiP simulations is shown in (e) and (f). The IOT frequencies (histograms) for contrail cirrus and contrails are shown in (d) and (g), respectively.**

In general, the model results demonstrate increased consistency with MSG observations when $q_{ANN}$ is incorporated in CoCiP. Given that ice supersaturation typically exhibits large horizontal but shallow vertical extensions (Spichtinger et al., 2003), a minor adjustment in cruising altitude, avoiding regions of high humidity, can potentially reduce contrail radiative forcing. An improved representation of humidity thanks to an ANN approach is thus crucial for more accurate predictions of the contrail cirrus radiative effect.

## 6 Summary and conclusions

The distribution of relative humidity in the UTLS from NWP models, which plays a vital role in the parameterizations of natural cirrus and contrail cirrus properties, is subject to large uncertainties. In this study we propose a humidity bias correction method for relative humidity from ERA5, $RHi_{ERA5}$, particularly for ice supersaturation in the UT, using an ANN technique.



The novelty of this study lies in the incorporation of thermodynamic and dynamical atmospheric quantities for the given time and height together with atmospheric properties from previous times and nearby altitudes. The atmospheric humidity improvement method consists of an ANN developed using atmospheric variables from ERA5, along with collocated
measurements of water vapor from IAGOS. The ERA5 data includes temporal and vertical dependencies of humidity on meteorological conditions, combining not only the historic data (-6 h, -2 h) and current time but also ±2 ERA5 pressure layers around the flight latitude of IAGOS. The target region covers the Atlantic, Europe and Africa, spanning 0 to 80°N and 30°W to 50°E and pressure levels from 400 to 200 hPa.

The analysis of biases between collocated $RHi_{ERA5}$ and $RHi_{IAGOS}$ reveals an underestimation of $RHi_{ERA5}$ within the UT and
an artificial RHi occurrence peak near 100% due to the cloud saturation adjustment by ECMWF NWP. The ERA5-IAGOS collocated data is processed and the variables for training the ANN model for humidity correction are selected based on the discussion of the temporal evolution of meteorological variables. Humidity, temperature, and geopotential (as a variable for the altitude) have a main impact on the $RHi_{ANN}$ results, while other meteorological variables, including vertical velocity, divergence, horizontal wind speed, and relative vorticity, have a minor but measurable influence.

Using this ANN humidity correction, the MAE of $RHi_{ANN}$ when comparing to $RHi_{ERA5}$ both against $RHi_{IAGOS}$ is reduced from 15.74% to 6.62%, 16.24% to 6.77%, 11.19% to 4.55%, and 9.75% to 4.33%, in all sky UT, cloudy UTLS, clear UTLS, and all sky LS regions, respectively, presenting remarkable improvements, particularly in the all sky UT and cloudy UTLS regions. A previously existing artificial occurrence peak at RHi = 100 % in $RHi_{ERA5}$, which is caused by the cloud saturation adjustment in NWP, has been removed completely by the ANN.

The representation of ice supersaturation in $RHi_{ERA5}$ and $RHi_{ANN}$ with respect to $RHi_{IAGOS}$ was assessed with the calculation of the ETS value. The dynamic-based humidity correction leads to an increase in ETS from 0.22, 0.25, and 0.14 (ERA5) to 0.64, 0.65, and 0.65 by the ANN, respectively, in the all sky UT, cloudy UTLS, and all sky LS regions. The skill of ice supersaturation prediction improves considerably.

The forecast of optical and radiative properties of cirrus and contrail cirrus, based on the ANN humidity correction, is
exemplarily assessed with CoCiP simulations using IFS weather data and the ANN corrected data and MSG satellite observations for one case between 35°N and 60°N (over the NAR) at 10:00 UTC on 14 April 2021. The result shows better agreement in ice optical thickness between model simulations with humidity correction and satellite observations in this contrail situation.

Teoh et al. (2022) and Wolf et al. (2023) utilize IAGOS measurements to correct $RHi_{ERA5}$ with statistical methods. Our study
shows the potential of the emerging field of machine learning-based weather prediction post-processing, in which forecast outputs are improved using historical observations and analysis data. How the current atmospheric states influence the future development of humidity patterns has been highlighted. One issue in the existing model data, where the frequency and degree of ice supersaturation in the UT are consistently underestimated due to the practice of the cloud saturation adjustment has been successfully addressed by the ANN model. The method demonstrates competitive performance, as seen by the decreased MAE
and larger ETS compared to the accuracy of the aforementioned statistical methods.



Incorporating more water vapor data from the fleet-wide observations of humidity within the UTLS can further improve this method. Our findings suggest potential applications for aircraft diversion strategies to avoid ice supersaturation regions and reduce contrail cirrus climate impact. Further research on applying humidity correction methods to weather forecasts is vital for improving our understanding of the global cloud radiation budget. Our improved humidity predictions could serve as
benchmarks for the measurements of further aircraft campaigns, as the assimilation or reference data set for NWP or climate models for a better parameterization of ice supersaturation. The method could also be applied to other weather forecast models, including those from ECMWF and national Weather Services. Additionally, increased resolution of NWP models at the tropopause is required for better cirrus and contrail forecasting. Combining modeled meteorological conditions and their temporal changes, along with measured humidity from long-term data sets, is essential for a more realistic representation of
RHi and the subsequent processes like cloud and contrail formation in the UTLS, and their climate impact.

*Data availability*

IAGOS measurements are available at https://iagos.aeris-data.fr/ (IAGOS database). The ERA5 and IFS atmospheric profiles are obtained from Climate Data Store (https://cds.climate.copernicus.eu/) or ECMWF directly. The SEVIRI data are provided
by EUMETSAT (European Organisation for the Exploitation of Meteorological Satellites). The CoCiP model code can be accessed from https://py.contrails.org/install.html. The machine learning technique implementation is based on the open-source platform TensorFlow (https://www.tensorflow.org). The required software packages are Python (https://www.python.org), Keras (https://pypi.org/project/keras/), and Scikit-learn (https://pypi.org/project/scikit-learn/).

*Author contributions*

ZW conceived the study concept, developed the methods, and wrote the paper. LB and CV advised the study and provided feedback on the paper. KG contributed to explanations of the dependence of humidity on meteorological conditions. MIH contributed to the initial method and ERA5 data evaluation. SR and AP coordinated the IAGOS research infrastructure and provided the data. SK helped with the interpretation of humidity bias in ERA5. All authors contributed to and commented on
the paper.

*Competing interests*

One of the (co-)authors is a member of the editorial board of *Atmospheric Chemistry and Physics*.


*Acknowledgements*

We thank the IAGOS European Research Infrastructure for excellent global aircraft measurements and ECMWF and EUMETSAT for providing the modeled atmospheric data and MSG/SEVIRI observations. ZW was supported by the DLR (Deutsches Zentrum für Luft- und Raumfahrt) / DAAD (Deutscher Akademischer Austauschdienst) Research Fellowships –



Doctoral Studies in Germany, 2020 under grant no 57540125 and now by the LUFO project MEFKON. CV, LB and SK are supported by the Deutsche Forschungsgemeinschaft (DFG, German Research Foundation) within SPP-1294 HALO under project no VO1504/9-1; 522359172, and TRR 301 – Project ID 428312742, and by European Union CONCERTO and by SESAR JU CICONIA. IAGOS data were created with support from the European Commission, national agencies in Germany (BMBF), France (MESR), and the UK (NERC), and the IAGOS member institutions (http://www.iagos.org/partners). The

participating airlines (in the year 2020: Deutsche Lufthansa, Air France, China Airlines, Hawaiian Airlines) supported IAGOS by carrying the measurement equipment free of charge. The data are available at http://www.iagos.fr thanks to additional support from AERIS. We thank Andreas Schäfler from DLR for valuable discussions.

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
