# Peer review of "Machine learning for improvement of upper tropospheric relative humidity in ERA5 weather model data"

_EGUsphere, 2024_

## Referee Comment (RC1)

Review of "Machine learning for improvement of upper tropospheric relative humidity in ERA5 weather model data", by Wang and coauthors, EGUSphere-2024-2012.

The goal of this study is to improve the estimations of the relative humidity with respect to ice in the upper troposphere/lower stratosphere from ERA5 fields using machine learning. To achieve this goal, the authors develop an artificial neural network model to correct relative humidity from ERA5, using thermodynamic conditions and dynamical quantities from ERA5, together with water vapor data from IAGOS commercial aircraft measurements. The model RHi is trained using these data. Overall, this is an excellent article, and, although I spent a lot of time thinking about different aspects of the model, I have relatively few comments.

Line 64. You may want to mention that calibration of RH instruments at temperatures below $0^0$C are difficult, with increasing difficulty at decreasing temperatures.

145. A brief mention of how ciwc is derived from ERA5, as it is so important a variable to differentiate cloudy and cloud-free regions, would be helpful.

168. "accounts for" rather than "copes".

189. At temperatures below $-40^0$C, $RH_i$ should not exceed 100%, even though it may be above that value for a short period of time.

202. contrail formation threshold. A sentence is needed here-how is the threshold met? The Schmidt-Appleman criteria?

217-218. What is the spatial resolution of the radiometer?

Section 4.  Model evaluation and results

An additional evaluation could be made by using collocated CALIPSO lidar data.

Figure 5. You get cloudy skies from ERA5 when IWC is indicated? How accurate is this?

Figures 4, 5, 6. Could you possibly partition by $5^0$C increments in supplemental information to see if RHi>100% at temperatures below -40C, because it shouldn't.

408. You may want to mention that ECLIF3 was an experiment involving synthetic aviation fuel. The DLR Falcon jet research aircraft participated in the project, making some measurements simultaneously with the A350 aircraft. Could you use the ANN model in these situations as a test of the model?

---

## Author Response (AR2)

**Iteration: Correction**

**Responses to editor**

We thank the editor Annika Oertel for her positive judgement on the manuscript and the acceptance for publication after helpful technical corrections.

In the following we number the editor`s comments (EC) and reply (R) to them individually.

Dear authors,

We thank you for carefully addressing the reviewer and community comments. Following technical corrections, your manuscript is accepted for publication.

Additional reviewer comments:

EC1: Just one points. Figure A7. The rectangles I assume are the HALO aircraft track. That should be put in the figure caption.

R1a) author's response

Thank you for your comments. I think you are referring to Figure 7 in the main text. The selected case at 10 UTC on 14 April 2021 corresponds to the ECLIF3 campaign, during which the DLR Falcon 20 research aircraft conducted the chase flight to measure aircraft emissions. A comparison with aircraft measurements will be performed in a future study, and the aircraft track has not been analyzed in this manuscript. As you suggested, the linear structures of contrails correspond to the associated flight tracks of daily commercial flights, and we have included this description in the caption. Or if you mean Figure A7 in the response, we have included it in the caption as well.

R1b) manuscript changes

Captain of Figure 7: … "The linear structures of contrails correspond to the associated flight tracks." …

EC2: If the authors can find the time, I would suggest removing the map background on figure S6, which does not really serve a purpose but makes the data lines harder to read.

R2a) author's response

Thank you for your suggestions. We agree that the map background in Figure S6 was excessive and have removed it in the final version of the manuscript, updating Figure S6 accordingly.

R2b) manuscript changes

[Figure]

Figure A1 (S6): RHi derived from (a) ERA5 or (c) the ANN model and the differences relative to AIMS measurements in (b) and (d) obtained from the HALO aircraft on 21 July 2021 during the CIRRUS-HL campaign. The lines present the HALO flight track.

Best regards,

Annika Oertel

**Changes to the names of the Forschungszentrum Jülich institutes and adjustments to the affiliations:**

The Forschungszentrum Jülich has internally updated the names of its institutes. Consequently, the related authors Michaela I. Hegglin[3, 4], Susanne Rohs[5], and Andreas Petzold[5] have not moved to new affiliations but have updated the names of their affiliations to reflect these changes.

[3]Institute of Energy and Climate Systems 4 – Stratosphere (ICE-4), Forschungszentrum Jülich, Jülich, 52428, Germany

[4]Department of Meteorology, University of Reading, Reading, RG6 6ET, United Kingdom

[5]Institute of Energy and Climate Systems 3 – Troposphere (ICE-3), Forschungszentrum Jülich, Jülich, 52428, Germany

**Iteration: Revised submission**

**Response to anonymous reviewer #1**

We thank the reviewer for his/her positive judgement on the manuscript and the helpful comments, which we address in this revision. (1) We have provided more detailed explanations on distinguishing cloudy and clear-sky regions, the derivation and threshold of IWC in ERA5, and the relationship between RHi and temperature from both the ERA5 model and measurements. (2) We further explain for using CIRRUS-HL campaign measurements but no other data for model validations. To this end, we have included clear explanations regarding contrail formation threshold and satellite observation resolution.

In the following we enumerate the referee's comments (RC) and our replies (R) to each, referencing the corresponding tracked changes in the manuscript.

**RC:** Summary of paper:

Review of "Machine learning for improvement of upper tropospheric relative humidity in ERA5 weather model data", by Wang and coauthors, EGUSphere-2024-2012.

The goal of this study is to improve the estimations of the relative humidity with respect to ice in the upper troposphere/lower stratosphere from ERA5 fields using machine learning. To achieve this goal, the authors develop an artificial neural network model to correct relative humidity from ERA5, using thermodynamic conditions and dynamical quantities from ERA5, together with water vapor data from IAGOS commercial aircraft measurements. The model RHi is trained using these data. Overall, this is an excellent article, and, although I spent a lot of time thinking about different aspects of the model, I have relatively few comments.

Comments:

**RC1:** Line 64. You may want to mention that calibration of RH instruments at temperatures below 0° C are difficult, with increasing difficulty at decreasing temperatures.

**R1a)** author's response

Thanks for pointing out this aspect. We have included the suggested explanation of the RH instrument calibration issue, as their measurements are a key source for assimilation in NWP models.

**R1b)** manuscript changes

L65-66: "While as the primary data source for the assimilation system, calibrating RH instruments at temperatures below 0°C is challenging, with the difficulty increasing as temperatures drop further."

**RC2:** 145. A brief mention of how ciwc is derived from ERA5, as it is so important a variable to differentiate cloudy and cloud-free regions, would be helpful.

**R2a):** author's response

Thank you for your comment, which indeed warrants further explanation. Cloud fraction and cloud ice/water content in ERA5 are derived from the prognostic cloud scheme. The text has been revised, and the relevant references have been added.

**R2b)** manuscript changes

L151-153: "*ciwc* represents the mass of cloud ice particles per kilogram of moist air, averaged over a grid box. It is estimated using the prognostic equations of the cloud scheme (Tiedtke, 1993; Forbes and Tompkins, 2011; Forbes and Ahlgrimm, 2014), which account for cloud ice growth through deposition."

References

Forbes, R. M. and Ahlgrimm, M.: On the representation of high-latitude boundary layer mixed-phase cloud in the ECMWF global model, Mon. Weather Rev., 142, 3425–3445, https://doi.org/10.1175/MWR-D-13-00325.1, 2014.

Forbes, R. and Tompkins, A.: An improved representation of cloud and precipitation, Tech. Rep., European Center for Medium-Range Weather Forecasting, https://doi.org/10.21957/nfgulzhe, 2011.

Tiedtke, M.: Representation of clouds in large-scale models, Mon. Weather Rev., 121, 3040–3061, https://doi.org/10.1175/1520-0493(1993)121<3040:ROCILS>2.0.CO;2, 1993.

**RC3:** 168. "accounts for" rather than "copes"

**R3a):** author's response

Replaced "copes with" with "accounts for" in L177.

**RC4:** 189. At temperatures below -40° C, RHi should not exceed 100%, even though it may be above that value for a short period of time.

**R4a)** author's response

Thank you for your question regarding data validity. As you pointed out, RHi can exceed 100% even when temperatures (T) are below -40 °C for some time, due to the ice supersaturation required for cirrus ice crystals formation and other non-equilibrium processes in the atmosphere. Long-term observations show that there is plenty of ice supersaturation in the upper troposphere, where T are usually colder than -40° C. Therefore, we think there is no need to constrain the measurement range to T above -40°, as we are also interested in the typical cirrus regime where T <-40°.

**RC5:** 202. contrail formation threshold. A sentence is needed here-how is the threshold met? The Schmidt-Appleman criteria?

**R5a):** author's response

Thank you for your comment. Contrails form when hot, humid exhaust mixes with colder ambient air below the critical temperature known as the Schmidt-Appleman criterion. The text has been revised to explain how this threshold is reached, and the relevant references have been added.

**R5b)** manuscript changes

L211-212: "When the ambient temperatures fall below the Schmidt-Appleman criterion threshold (Schumann, 1996) at two successive flight waypoints" …

Reference

Schumann, U.: On conditions for contrail formation from aircraft exhausts, Meteorol. Z., 5, 4–23, https://doi.org/10.1127/metz/5/1996/4, 1996.

**RC6:** 217-218. What is the spatial resolution of the radiometer?

**R6a)** author's response

Thank you for pointing out this detail. SEVIRI provides observations with a 3 km sampling distance at nadir. The text has been revised accordingly.

**RC7:** Section 4. Model evaluation and results. An additional evaluation could be made by using collocated CALIPSO lidar data.

**R7a)** author's response

Thank you for suggesting this approach to evaluate the improved RHi data with lidar data. Unfortunately, CALIPSO lidar provides cirrus optical thickness and other properties based on backscatter measurements, but it does not provide humidity data. Humidity retrieval requires temperature profiles, which are typically derived from weather model data. While AIRS satellite observations include water vapor measurements, they have coarse spatial resolution in the vertical direction with large altitude bins. Therefore, we refrain from an additional evaluation using for instance CALIPSO lidar data.

**RC8:** Figure 5. You get cloudy skies from ERA5 when IWC is indicated? How accurate is this?

**R8a)** author's response

Thank you for your questions. In this study, we use ice water content (IWC) to differentiate between cirrus clouds and cirrus-free regions within the ERA5 grid. For the IWC detection threshold, we reviewed the literature and found that while a threshold of 0.1 g/m³ is commonly used (e.g., Krämer et al., 2020), lower IWC values are also observed in cirrus cloud regions (see Table 1 in Hong and Liu, 2015). Therefore, we use a valid IWC (>0) as an indicator for cirrus cloud regions and acknowledge that different IWC thresholds could be applied for finer cirrus classifications, as discussed in the literature.

Since this analysis relies on ERA5 data, it reflects model cloudiness rather than direct observations. We chose ERA5 IWC data because ERA5 include all necessary input quantities of our ANN models, such as RHi and atmospheric physics parameters. Although potential discrepancies between model and real clouds may exist, this approach agrees with the scope of our current research, and the mismatch can be further explored in future studies.

**R8b)** manuscript changes

L183-184: hence we differentiate between "model" clear (cloudy) conditions using *ciwc* equal to zero for all the current and ±2 pressure layers from ERA5…

Reference

Hong, Y. and Liu, G.: The characteristics of ice cloud properties derived from CloudSat and CALIPSO measurements, J. Climate, 28, 3880–3901, https://doi.org/10.1175/JCLI-D-14-00666.1, 2015.

Krämer, M., Rolf, C., Spelten, N., Afchine, A., Fahey, D., Jensen, E., Khaykin, S., Kuhn, T., Lawson, P., Lykov, A., Pan, L. L., Riese, M., Rollins, A., Stroh, F., Thornberry, T., Wolf, V., Woods, S., Spichtinger, P., Quaas, J., and Sourdeval, O.: A microphysics guide to cirrus – Part 2: Climatologies of clouds and humidity from observations, Atmos. Chem. Phys., 20, 12569–12608, https://doi.org/10.5194/acp-20-12569-2020, 2020.

**RC9:** Figures 4, 5, 6. Could you possibly partition by 5° C increments in supplemental information to see if RHi>100% at temperatures below -40° C, because it shouldn't.

**R9a)** author's response

Thank you for your suggestions. The response provided here is the same as the one given to RC4.

**RC10:** 408. You may want to mention that ECLIF3 was an experiment involving synthetic aviation fuel. The DLR Falcon jet research aircraft participated in the project, making some measurements simultaneously with the A350 aircraft. Could you use the ANN model in these situations as a test of the model?

**R10a)** author's response

Thank you for highlighting this potential comparison. In Figure S6 of the supplement, we present RHi derived from ERA5 and the ANN model, along with the differences relative to AIMS measurement obtained from the HALO aircraft on July 21, 2021, during the CIRRUS-HL campaign. The CIRRUS-HL experiment involved the HALO research aircraft, satellites, and models to provide new insights into the effects of aviation on clouds and the climate impact of contrails over Central Europe and the North Atlantic flight corridor. These aircraft measurements are extensively used for model validation related to water vapor and are also utilized to improve humidity assimilation in weather forecasting models. I agree with your point that ECLIF3 focuses on measurements related to sustainable aviation fuel, which is somehow beyond the scope of model validation in this context.

**Responses to anonymous reviewer #2**

We thank the referee for highlighting the importance of this study and for helpful advice and constructive comments about our paper. The suggestions have led to a revised manuscript with two main goals: (1) While we confirm the minimal autocorrelation in the IAGOS data, we have reconstructed the training, validation and test datasets to ensure that model building and testing do not occur on the same day or overlap in any way. We now give additional details on the selection of the study region, considering humidity uplift from lower atmospheric levels, and describe the RHi distribution within the dataset, as well as provide further information on the development procedure of the algorithm. (2) We offer further insights into humidity, dynamics and temporal dependence, along with updated contributions of input variables to humidity prediction, and refined explanations regarding the importance of dynamical variables in RHi prediction. Additionally, we have included additional maps presenting humidity patterns. To this end, we have written concise explanations and modified pictures accordingly.

In the following we enumerate the referee's comments (RC) and our replies (R) to each, referencing the corresponding tracked changes in the manuscript.

**RC:** Summary of paper:

In this study, the authors train an artificial neural network to predict the distribution of relative humidity over ice in the UTLS over Western Europe. The network is trained on a mixture of thermodynamical and dynamical variables, although the former explain most of the prediction skill. The network is better than ERA5 at predicting RHi, and its inputs lead to better contrail prediction from the Cocip model in one case study.

The paper deals with an important topic. It is very well written. The introduction is excellent. The figures illustrate the discussion well, although I would have preferred to see more maps because scatterplots only give an incomplete indication of the ability of the network to reproduce patterns of humidity.

Others have commented in the online discussion on the need to better separate the training dataset from the validation dataset. I will not elaborate further on that aspect but revisions to the method are clearly needed there.

**Ra)** author's response

Thank you for the positive feedback on our proposed RHi improvement method. To better illustrate the accurate and enhanced spatial variability of RHi patterns over Western Europe, in addition to presenting the RHi distribution in the original Figure 6 of the manuscript, we have also added equidistant latitude-longitude maps from ERA5 and ANN RHi data at 200 hPa, taken at 08:00 UTC on July 21, 2021, corresponding to the flight of the HALO aircraft during the CIRRUS-HL campaign on that date.

Refer to Figure A1 below and Figure S5 in the revised supplement.

[Figure]

Figure A1 (S5): Patterns of (a) $RHi_{ERA5}$ and (b) $RHi_{ANN}$ at 200 hPa at 08:00 UTC on 21 July 2021.

The online discussion highlighted the needs to separate the training and validation datasets due to the autocorrelation present in the IAGOS data within a single ERA5 grid box during the IAGOS and ERA5 collocation. We appreciate this feedback and calculated the autocorrelation function of the IAGOS data series following the method of Dotzek and Gierens (2008). The results, displayed in Figure A2 here, show that autocorrelation reaches up to 0.04 at the original resolution (4 s, 1 km) before gradually declining to near zero at a resolution of 0.25 degrees (25 km), which corresponds to the ERA5 grid box size. It can be explained by the nature that the water vapor field is quite chaotic with steep gradients. Diao et al (2014, 2015) showed that in-situ RHi measurements taken with 1 Hz or 10 Hz instruments reveal small-scale structures in the RHi time series, in agreement with our finding. Therefore, we assume that autocorrelation can be disregarded for the averaged IAGOS RHi measurements within the ERA5 grid box.

[Figure]

Figure A2: Autocorrelation function of the IAGOS measurement series (4s, 1km).

In response, we have still revised our methodology to ensure that the training set consists of data from time periods that do not overlap with those used for either validation and testing. Specifically, we are now using 4 days of data for building the ANN model, followed by a 1-day gap, and then 1 day of data for validation or testing. The complexity of the ANN model has also been increased to better match the growing complicacy of the training data, including the addition of hidden layers and neurons, as well as weight initialization and batch normalization for each layer to enhance generalization. Revisions to the text, along with updated Figures 4, 5, 6, 7, and S6 for RHi using the new models, are provided below. Updates on the validation of $T_{IAGOS}$ and $q$ predictions are presented in Figures S1, S3, and S4 in the revised supplement.

**Rb)** manuscript changes

L174-176: "To ensure that the training set consists of data from time periods that do not overlap with those used for either validation and testing, we now use 4 days of data to build the ANN model, followed by a 1-day gap, and then 1 day of data for validation or testing."

L291-294: "In Sect. 2.3, four consecutive days of samples from the ERA5-IAGOS collection are allocated for model training, with the following day excluded to avoid overlap with the continuous weather system, and another day reserved either for validation, to evaluate the model's generalization to unseen data during training, or for testing."

L43-45 in the supplement: To ensure model robustness "and construct an independent test data set, we now use a sequence-based split: four consecutive days of data are used to build the ANN model, followed by a 1-day gap, with the subsequent day's data reserved for validation or testing".

L286-287: "We use 3 hidden layers, each with 100 neurons and He weight initializer (He et al., 2015), along with batch normalization between layers to improve generalization. The humidity output is referred to as $RHi_{ANN}$."

L298: "an ANN for $q$ is implemented, with 300 neurons in each hidden layer and…"

L18-20 in the Abstract: "The ANN shows excellent performance and the predicted RHi in the UT has a mean absolute error MAE of 5.7% and a coefficient of determination R² of 0.95, which is significantly improved compared to ERA5 RHi (MAE of 15.8%; R² of 0.66)."

L348-350: "The MAE decreases significantly from 15.82% (ERA5) to 5.71% (ANN), the $R^2$ values increase from 0.66 (ERA5) to 0.95 (ANN), and the root mean spare error RMSE decreases from 20.52% (ERA5) to 7.88% (ANN)."

[Figure]

Figure A3 (4): Distribution of (a) $RHi_{ERA5}$ and (b) $RHi_{ANN}$ versus $RHi_{IAGOS}$ in the UT in all sky (clear and cloudy conditions) in the test data set. The number of data sets N, the mean absolute error MAE and the coefficient of determination R² are shown in the panels.

L354-356: "In the cloudy (Fig. 5a-b) and clear sky (Fig. 5c-d) conditions in the UTLS, the MAE of the RHi decreases from 16.28% (11.21%) to 5.95% (4.28%), respectively. Also, the R² increases by 0.30 (0.23) to 0.95 (0.95) for the two scenarios."

L361-363: "The ANN model also has strong skills of RHi correction in the LS, see Fig. 5e and f. R² values increase from 0.59 (ERA5) to 0.95 (ANN), similar to the UT region in Fig. 4. The improvement of RHi prediction by the ANN is also documented by the decrease of MAE by 6.07%."

L377: "with a MAE of approximately 5.8%..."

L485-486: "Using this ANN humidity correction, the MAE of $RHi_{ANN}$ when comparing to $RHi_{ERA5}$ both against $RHi_{IAGOS}$ is reduced from 15.82% to 5.71%, 16.28% to 5.95%, 11.21% to 4.28%, and 9.78% to 3.71%..."

[Figure]

Figure A4 (5): Comparison of $RHi_{ERA5}$ (left column) and $RHi_{ANN}$ (right column) against $RHi_{IAGOS}$ in the (a) and (b) cloudy UTLS, (c) and (d) clear sky UTLS, and (e) and (f) clear sky and cloudy (or all sky) LS regions in the test data set. The number of data sets N, the mean absolute error MAE and the coefficient of determination R² are indicated in the individual panels.

L369-371: "As opposed to this, Fig. 6d shows that $RHi_{ANN}$ and $RHi_{IAGOS}$ have a closer agreement, with an MBE of about ±11% for all UT measurements up to 140%. The RHi between 80% and 130% in the important range for cirrus clouds is well represented by the ANN with an MBE better than +/- 7%."

[Figure]

Figure A5 (6): Frequency distribution (a and c) and overall mean biased error MBE (%) (b and d) of $RHi_{ERA5}$ and $RHi_{ANN}$ against $RHi_{IAGOS}$ in the clear and cloudy UT (grey) in the test data set.

[Figure]

Figure A6 (7): Distributions of IOT for contrails and cirrus retrieved from (a) MSG observations using the CiPS algorithm, and simulated using the CoCiP model with (b) $q_{ANN}$ or (c) $q_{IFS}$ at 10:00 UTC on 14 April 2021. The IOT distribution for contrails from CoCiP simulations is shown in (e) and (f). The linear structures of contrails correspond to the associated flight tracks. The IOT frequencies (histograms) for contrail cirrus and contrails are shown in (d) and (g), respectively.

[Figure]

Figure A7 (S6): RHi derived from (a) ERA5 or (c) the ANN model and the differences relative to AIMS measurements in (b) and (d) obtained from the HALO aircraft on 21 July 2021 during the CIRRUS-HL campaign. The lines present the HALO flight track.

References

Diao, M., Zondlo, M. A., Heymsfield, A. J., Avallone, L. M., Paige, M. E., Beaton, S. P., Campos, T., and Rogers, D. C.: Cloud-scale ice-supersaturated regions spatially correlate with high water vapor heterogeneities, Atmos. Chem. Phys., 14, 2639–2656, https://doi.org/10.5194/acp-14-2639-2014, 2014.

Diao, M., Jensen, J. B., Pan, L. L., Homeyer, C. R., Honomichl, S., Bresch, J. F., and Bansemer, A.: Distributions of ice supersaturation and ice crystals from airborne observations in relation to upper tropospheric dynamical boundaries, J. Geophys. Res.-Atmos., 120, 5101–5121, https://doi.org/10.1002/2015JD023139, 2015.

Dotzek, N. and Gierens, K.: Instantaneous fluctuations of temperature and moisture in the upper troposphere and tropopause region. Part 2: Structure functions and intermittency, Meteorol. Z., 17, 323–337, https://doi.org/10.1127/0941-2948/2008/0292, 2008.

He, K., Zhang, X., Ren, S., and Sun, J.: Delving Deep into Rectifiers: Surpassing Human-Level Performance on ImageNet Classification, 2015 IEEE International Conference on Computer Vision (ICCV), Santiago, Chile, 1026-1034, https://doi.org/10.1109/ICCV.2015.123, 2015.

I have a couple of additional comments that would need to be addressed before the study is published.

**RC1:** First, I am surprised by the selection of the study region, as shown in Figure 1. Why doesn't it extend further west? Given that the network relies on the temporal evolution of humidity it seems it would make sense to include the regions where most of the humid regions are either formed or advected from. There is plenty of IAGOS data over the North Atlantic and Eastern US too.

**R1a)** author's response

Thank you for highlighting this aspect. In selecting the study region, we aimed to account for both horizontal advection and vertical air mass transport, as these factors contribute significantly to the inclusion of humid regions. The moisture formed and transported from lower atmospheric levels to cruise altitude is also represented in our analysis. The distribution of $RHi_{IAGOS}$ is already included in Figure S2 of the supplement, showing that RHi density remains high, decreasing gradually from 100% and notably only as it approaches 150%, a trend consistent with findings in Wolf et al. (2023, Fig. 3) and Teoh et al. (2024, Fig. S4).

Our focus on the Eastern Atlantic, Europe, and Africa stems from the higher density of air traffic in these regions compared to others. This includes the concentrated morning eastbound and afternoon westbound transatlantic flights between the U.S. and Europe, which result in uneven sampling times over a 24-h period over the Atlantic Ocean (Schumann and Graf, 2012). Our regional selection enables us to establish more precisely the relationships between meteorological input variables and measured humidity across Europe, capturing temporal and regional variations effectively.

**R1b)** manuscript changes

L129-131: "We did not include the western regions of NAR, from where humid air is often advected, but instead focused on moisture originating and transported from lower atmospheric levels up to cruise altitude."

L136-138: "The distribution of $RHi_{IAGOS}$ shows high-density values that gradually decrease starting from 110% and drop significantly as they approach 150% (see Fig. S2 in the supplement), a trend consistent with findings in Wolf et al. (2023, Fig. 3) and Teoh et al. (2024, Fig. S4)."

References

Schumann, U. and Graf, K.: Aviation-induced cirrus and radiation changes at diurnal timescales, J. Geophys. Res.-Atmos., 118, 2404–2421, https://doi.org/10.1002/jgrd.50184, 2013.

Teoh, R., Engberg, Z., Schumann, U., Voigt, C., Shapiro, M., Rohs, S., and Stettler, M. E. J.: Global aviation contrail climate effects from 2019 to 2021, Atmos. Chem. Phys., 24, 6071–6093, https://doi.org/10.5194/acp-24-6071-2024, 2024.

Wolf, K., Bellouin, N., Boucher, O., Rohs, S., and Li, Y.: Correction of temperature and relative humidity biases in ERA5 by bivariate quantile mapping: Implications for contrail classification, EGUsphere [preprint], https://doi.org/10.5194/egusphere-2023-2356, 2023.

**RC2:** Second, the lack of importance of the dynamical variables in explaining the prediction is surprising. The explanation proposed by the authors, that of a strong correlation between thermodynamical and dynamical variables, is plausible. But time scales are crucial in that correlation, so I wonder whether the study design somehow maximises the correlation. By the choice of the study region for example, which excludes the North Atlantic where dynamics might affect the evolution of humidity more clearly? Or by the choice of lead times? On that point, the question on temporal dependence asked in Section 3.1 on lines 227-228 is never really answered. How much does including distributions 6hr before current time improve the network prediction, for example?

**R2a):** author's response

Thank you very much for your insightful questions and for raising the issue of the limited apparent importance of dynamical variables in the RHi prediction model. Based on further investigation, we now provide a clearer explanation of their impact.

1. Updated $K_x$ values for thermodynamic and dynamical variables in RHi prediction

After ensuring the separation of training, validation, and test data on different days, we recalculated the relative contributions of variables to the new prediction of RHi (denoted as $K_x$, equation (1) in the manuscript, not necessarily less than 1). Following the suggestion from community comment #4, we set the investigated ERA5 feature to its mean value while keeping the other variables unchanged for the computation of $K_x$. The importance of the dynamical variables - vertical velocity (*w*), divergence (*d*), and especially horizontal wind components (*u*, *v*) and relative vorticity (*vo*) - has now increased.

For instance, the $K_x$ values for *RHi*, temperature (*T*), and geopotential (*z*) remain high, at 1.85, 1.24, and 1.39. $K_x$ values for *w* and *d* have now risen to 0.29 and 0.26, respectively, and for *u*, *v*, and *vo* to 0.98, 0.75, and 0.74. We acknowledge that thermodynamic variables (e.g., *RHi*, *T*) may inherently capture some of the dynamical trends. Hofer et al. (2024) also shows that the predictor $RHi_{ERA5}$ has the greatest impact on humidity predictions, while the explanatory power of the dynamical proxies is insufficient when only using data from the current time and level. However, this updated analysis confirms that incorporating a broader vertical region and the historical (2 and 6h) times into the dynamical variables do indeed play a more prominent impact in the ANN model than previously thought, contributing to the understanding of humidity evolution. Revisions to the text and Figure A8 (updated Figure 3) are provided below.

2. Moisture uplift rather than advection from the North Atlantic

You rightly point out that our study region does not include the whole North Atlantic, where dynamical processes could have a clear influence on humidity evolution. As stated in response to your suggestion in RC1, we instead focused on moisture uplift from lower atmospheric levels rather than horizontal advection from the North Atlantic.

3. Temporal dependence of RHi on meteorological variables

As for the temporal dependence mentioned in Sect. 3.1, we now provide a more detailed analysis of the impact of past meteorological variables on RHi predictions. We have before calculated the Pearson correlation coefficients between thermodynamic and dynamical variable from ERA5 at various lead times (up to 24 hours prior) and measured $RHi_{IAGOS}$ at the time and location of IAGOS data acquisition, but wasn't explained in the discussion version of the manuscript.

Based on our calculations, for this 6-hour lag range, the correlation between $RHi_{ERA5}$ and $RHi_{IAGOS}$ decreases by about 5.4% from 0.49 from 6 hours prior to the current time, while $T_{ERA5}$ and z show constant significant correlations with $RHi_{IAGOS}$ (around -0.5 and 0.4). w consistently demonstrates negative correlations with RHi, with an absolute correlation decreases of 86% from -0.11 from the 6-hour lag to the current time. For horizontal winds (u, v), correlations fluctuate around 0.34 and 0.44, while d exhibits the higher correlation at the 6-hour lag, decreasing from 0.18 to the current time by about 83%. In contrast, vo continues to exhibit negative correlations with $RHi_{IAGOS}$, with an increasing absolute correlation coefficient that approaches -0.2. These results demonstrate the correlation between current RHi and meteorological conditions at preceding times and have been added as the new Sect. S2 in the revised supplement.

4. Effect of including lead times on model performance

We have tried before that including meteorological distributions from 6h before current time slightly improves the performance of the RHi prediction model, as seen in a reduction of both MAE (from 2.31% to 2.21%) and RMSE (from 4.01% to 3.64%) in the validation test. Additionally, we assessed the influences of introducing different time lags (1h, 2h, 3h, 6h), and observed that the degree of the MAE and RMSE decrease increases with larger time lags. We then chose to include data from current time, 2h, and 6h intervals to balance prediction accuracy and computational efficiency. These results, along with supporting metrics, are presented in Table A1 below and Table S1 in the new Sect. S2 in the supplement.

5. Literature about airmass transport on humidity evolution from trajectory analysis

We have also referred to the trajectory analysis in Dyroff et al. (2015) to indicate that the RHi bias is linked to air masses from approximately 230 hPa in high northern latitudes, likely affected by both vertical intrusions and horizontal transport. Thank you for your feedback, we now add more detailed explanations mentioned above.

Reference

Hofer, S., Gierens, K., and Rohs, S.: How well can persistent contrails be predicted? An update, Atmos. Chem. Phys., 24, 7911–7925, https://doi.org/10.5194/acp-24-7911-2024, 2024.

**R2b)** manuscript changes

L18 in the Abstract: "while other dynamical variables are of low to moderate or high importance."

L321-323: "they provide a moderate and non-negligible contribution to the accuracy of the RHi prediction model. In fact, *w* and *d* show $K_x$ of 0.29 and 0.26 and those for *u*, *v*, and *vo* even higher, which are 0.98, 0.75, and 0.74. There is generally less importance in the contributions of the variables representing dynamical quantities…"

L325-326: "The fact that dynamical variables for instance particularly u, v, vo are closely as important as $RHi_{ERA5}$, $T_{ERA5}$, and *z* for the description of the physical processes that lead to the decrease/increase of relative humidity in Sect. S2 in the supplement…"

L329-332: "Hofer et al. (2024) shows that $RHi_{ERA5}$ is the most influential predictor for humidity predictions, while the explanatory power of dynamical proxies is insufficient when only using data from the current time and level. However, our updated analysis confirms that incorporating a broader vertical region and the historical time into the dynamical variables has a more significant impact on the ANN model and contributes to the understanding of humidity evolution."

L483-484: "while other meteorological variables, including horizontal wind speed, relative vorticity, vertical velocity, and divergence, have a high or moderate to low but measurable influence."

[Figure]

Figure A8 (3): Relative importance of the individual variables to the ANN model for predicting RHi.

L254-256: "Further computations of the Pearson correlation coefficient between $RHi_{IAGOS}$ and temporal meteorological variables from ERA5, and the impact of including data distributions from hours before the current time on improving the network prediction, are explained in Sect. S2."

New Sect. S2 in the supplement: "The correlation between $RHi_{IAGOS}$ and ERA5 temporal meteorological variables".

"We have determined the temporal dependence of measured $RHi_{IAGOS}$ at the time and location of IAGOS data acquisition on meteorological variables at the preceding time up to 24 hour prior through the calculation of the Pearson correlation coefficient. Based on the calculations, compared to the time 6 hour before, the correlation of $RHi_{ERA5}$ and $RHi_{IAGOS}$ from 0.49 decreases by about 5.4% at the current time. The correlations for $T_{ERA5}$ and z with $RHi_{IAGOS}$ are also statistically significant and almost constant, with coefficients of about -0.5 and 0.4. *w* consistently demonstrates negative correlations with upward motion, resulting in cooling and an increase in RHi. The absolute correlation decreases from the 6-h time lags to

the current time from -0.11 by about 86%. The correlation for $u$ and $v$ tends to fluctuate around 0.34 and 0.44. $d$ generally exhibits positive correlations, with the highest value occurring around the 4-h to 5-h time lag, are 0.18 at the 6-h time lag higher than that of 0.03 at the current time by about 83%. In contrast, $vo$ continues to exhibit negative correlations with $RHi_{IAGOS}$, with an increasing absolute correlation coefficient that approaches -0.2.

Including meteorological data from 6 hour prior improves the accuracy of the RHi prediction model on the validation dataset, reducing the MAE from 2.31% to 2.21% and the RMSE from 4.01% to 3.64%. The effect of time lags on model accuracy is calculated and presented in Table S1. As meteorological variables from 1, 2, 3, and 6 hours before the current time are introduced, the decrease in MAE and RMSE gradually becomes more significant. To balance information richness with computational efficiency, we choose the combination of current time, 2 hour, and 6 hour."

Table A1 (S1): Impact of including data distributions from 6 hours prior on network prediction accuracy.

| Scenarios | MAE (%) | RMSE (%) | $R^2$ |
|---|---|---|---|
| current | 2.31 | 4.01 | 0.99 |
| current, -1 h | 2.21 | 4.17 | 0.98 |
| current, -2 h | 2.3 | 4.01 | 0.98 |
| current, -3 h | 2.33 | 4.01 | 0.98 |
| current, -6 h | 2.21 | 3.64 | 0.98 |
| current, -2h, -6h | 2.23 | 3.78 | 0.99 |

Other comments:

**RC3:** Lines 188-189: I am not sure that the RHi peak is always artificial. Sanogo et al. (2024) https://doi.org/10.5194/acp-24-5495-2024 suggest that the peak is seen in IAGOS in cloudy conditions. See their Figures 4 and 5

**R3a):** author's response

Thank you for pointing out this ambiguous description. The RHi peak reported by Sanogo et al. (2024) indeed reflects the maximum observed RHi values from IAGOS, as you suggested. Meanwhile, the RHi = 100% peak in ERA5 data mainly results from saturation adjustments inherent in numerical weather prediction models. We agree with your feedback and have revised "artificial" to "partially artificial" for clarity.

**R3b)** manuscript changes

L197-200: In addition, a "partially" artificial occurrence accumulation peak exists in the ERA5 data set at $RHi_{ERA5}$ = 100%. "In $RHi_{IAGOS}$, a small peak is observed between 100% and 110% under cloudy conditions (Sanogo et al., 2024). However, much of the accumulation peak in the ERA5 data is attributed to the cloud saturation adjustment in NWP models".

We have also removed terms such as "artificial" when describing the occurrence peak in the following text.

Reference

Sanogo, S., Boucher, O., Bellouin, N., Borella, A., Wolf, K., and Rohs, S.: Variability in the properties of the distribution of the relative humidity with respect to ice: implications for contrail formation, Atmos. Chem. Phys., 24, 5495–5511, https://doi.org/10.5194/acp-24-5495-2024, 2024.

**RC4:** Lines 308-311: Can you clarify how that statement relates to the statement on correlation made earlier in the paragraph?

**R4a)** author's response

Thank you for highlighting this ambiguous description. The statement about connections between ANN layers relates only to the calculation of importance metrics and not to the correlation between different meteorological variables mentioned earlier in the paragraph. We have removed the statement you pointed out and expanded the discussion on the impacts of dynamical variables on RHi predictions, incorporating insights from the literature and our analysis. The text has been revised accordingly for clarity, and please refer to our responses to RC2 for further details.

**Responses to community**

We thank the community for their helpful advice and constructive comments on our paper. Their suggestions have led to a revised manuscript focusing on three main objectives: (1) While the autocorrelation of IAGOS data is verified to be minimal, we have reconstructed the training, validation and test datasets to ensure no overlap or same-day usage between model building and testing. This approach supports the model's application in both retrospective analyses and real-time forecasting. (2) We included additional contingency table metrics beyond ETS to validate our improved ice supersaturation prediction (3) We also provide additional details on model construction and interpretation. For instance, the model inputs encompass meteorological variables across multiple conditions (current and prior times as well as surrounding pressure levels), normalization of input and output data during training, and a refined approach for calculating the relative importance of meteorological variables in RHi prediction by assessing changes in loss when each variable is set to its mean value. To this end, we have also added clear explanations and updated pictures accordingly.

**Response to community #1 Kevin McCloskey**

We thank Kevin McCloskey for highlighting the importance of the study and for helpful comments on further separating training set and validation/test sets before model construction, which we address in the revision of the manuscript.

In the following we number the community's comments (CC) and our replies (R) to each, referencing the corresponding tracked changes in the manuscript.

**CC:** Hello, this could be a very impactful finding if the ANN model generalizes well to weather conditions that it hasn't seen. I notice though in your Supplemental S2 section you describe randomly splitting the IAGOS waypoints into train/validation/test sets. Doing your cross validation in this way has a risk that your ANN model is overfitting. This is likely not a problem if you restrict your usage of the trained ANN to retrospective studies where the model inference is only applied to ERA5 data in the same times/places the model was trained on. However, if you attempted to apply an ANN trained in this way to a forecast of weather which has not happened in the real world yet, you would likely see a drop in metrics. To report metrics that are predictive of how the model will perform when applied to a weather forecast, it is best practice to train the ANN on an archived weather forecast (eg, ECMWF HRES) and use a chronological cross validation split: ie, the train set is comprised of data from time periods that are disjoint from the time periods used for the validation and test sets. For example, don't include in your validation/test sets any data from days that were included in your training set. This type of cross validation setup avoids the risk of the ANN 'memorizing' specific datapoints from the training set which are effectively also present in the validation/test sets, in a way that would not be the case when you apply the ANN to real weather forecast data. This is especially a concern here given the IAGOS waypoints occur once every 4 seconds and so adjacent datapoints (having extremely similar model inputs and target outputs) will frequently be randomly split across the train/test boundary. With the current cross validation setup, the impact of this model still seems strong, but limited to use in retrospective analyses.

**Ra)** author's response

Thank you for emphasizing the importance of this humidity prediction approach and acknowledging the potential limitation of using the trained ANN solely for retrospective studies, as it may be overfitted to the specific times and locations it was trained on, limiting its applicability for real-time weather forecasting.

In response, we have revised our methodology to ensure that the training set consists of data from time periods that do not overlap with those used for validation and testing. Here, we continue to use ERA5-IAGOS collocation data rather than ECMWF HRES, as ERA5 provides greater accuracy by incorporating past observational data. Referring to the cross-validation setup, specifically, we are now using 4 days of data for building the ANN model, followed by a 1-day gap, and then 1 day of data for either validation or testing. The complexity of the ANN model has also been increased to better match the growing complicacy of the training data, including the addition of hidden layers and neurons, as well as weight initialization and batch normalization for each layer to enhance generalization. Revisions to the text, along with updated Figures 4, 5, 6, 7, and S6 for RHi using the new models, are provided below. Updates on the validation of $T_{IAGOS}$ and $q$ predictions are presented in Figures S1, S3, and S4 in the revised supplement.

The ANN, trained in this way, was further tested on real weather forecast data from a separate test set, showing no drop in MAE and R² metrics thanks to the refinement of the ANN model settings.

In response to your concerns that adjacent IAGOS data points could result in highly similar model inputs and target outputs for the ANN, we calculated the autocorrelation function of the IAGOS data series following the method of Dotzek and Gierens (2008). The results, displayed in Figure A1 below, show that autocorrelation reaches up to 0.04 at the original resolution (4 s, 1 km) before gradually declining to near zero at a resolution of 0.25 degrees (25 km), which corresponds to the ERA5 grid box size. It can be explained by the nature that the water vapor field is quite chaotic with steep gradients. Diao et al (2014, 2015) showed that in-situ RHi measurements taken with 1 Hz or 10 Hz instruments reveal small-scale structures in the RHi time series, in agreement with our finding. Therefore, we assume that autocorrelation can be disregarded for the averaged IAGOS RHi measurements within the ERA5 grid box. However, we have revised our method according to your suggestions as described before.

[Figure]

Figure A1: Autocorrelation function of the IAGOS measurement series (4s, 1km).

Based on the suggested cross-validation setup, test results, and statistical evidence of minimal autocorrelation in the input and output data, we believe the model's effectiveness remains robust. This approach supports its application not only in retrospective analyses but also in real-time forecasting.

**Rb)** manuscript changes

L174-176: "To ensure that the training set consists of data from time periods that do not overlap with those used for either validation and testing, we now use 4 days of data to build the ANN model, followed by a 1-day gap, and then 1 day of data for validation or testing."

L291-294: "In Sect. 2.3, four consecutive days of samples from the ERA5-IAGOS collection are allocated for model training, with the following day excluded to avoid overlap with the continuous weather system, and another day reserved either for validation, to evaluate the model's generalization to unseen data during training, or for testing."

L43-45 in the supplement: To ensure model robustness "and construct an independent test data set, we now use a sequence-based split: four consecutive days of data are used to build the ANN model, followed by a 1-day gap, with the subsequent day's data reserved for validation or testing".

L286-287: "We use 3 hidden layers, each with 100 neurons and He weight initializer (He et al., 2015), along with batch normalization between layers to improve generalization. The humidity output is referred to as $RHi_{ANN}$."

L298: "an ANN for $q$ is implemented, with 300 neurons in each hidden layer and..."

L18-20 in the Abstract: "The ANN shows excellent performance and the predicted RHi in the UT has a mean absolute error MAE of 5.7% and a coefficient of determination R² of 0.95, which is significantly improved compared to ERA5 RHi (MAE of 15.8%; R² of 0.66)."

L348-350: "The MAE decreases significantly from 15.82% (ERA5) to 5.71% (ANN), the $R^2$ values increase from 0.66 (ERA5) to 0.95 (ANN), and the root mean spare error RMSE decreases from 20.52% (ERA5) to 7.88% (ANN)."

[Figure]

Figure A2 (4): Distribution of (a) $RHi_{ERA5}$ and (b) $RHi_{ANN}$ versus $RHi_{IAGOS}$ in the UT in all sky (clear and cloudy conditions) in the test data set. The number of data sets N, the mean absolute error MAE and the coefficient of determination R² are shown in the panels.

L354-356: "In the cloudy (Fig. 5a-b) and clear sky (Fig. 5c-d) conditions in the UTLS, the MAE of the RHi decreases from 16.28% (11.21%) to 5.95% (4.28%), respectively. Also, the R² increases by 0.30 (0.23) to 0.95 (0.95) for the two scenarios."

L361-363: "The ANN model also has strong skills of RHi correction in the LS, see Fig. 5e and f. $R^2$ values increase from 0.59 (ERA5) to 0.95 (ANN), similar to the UT region in Fig. 4. The improvement of RHi prediction by the ANN is also documented by the decrease of MAE by 6.07%."

L377: "with a MAE of approximately 5.8%..."

L485-486: "Using this ANN humidity correction, the MAE of $RHi_{ANN}$ when comparing to $RHi_{ERA5}$ both against $RHi_{IAGOS}$ is reduced from 15.82% to 5.71%, 16.28% to 5.95%, 11.21% to 4.28%, and 9.78% to 3.71%..."

[Figure]

Figure A3 (5): Comparison of $RHi_{ERA5}$ (left column) and $RHi_{ANN}$ (right column) against $RHi_{IAGOS}$ in the (a) and (b) cloudy UTLS, (c) and (d) clear sky UTLS, and (e) and (f) clear sky and cloudy (or all sky) LS regions in the test data set. The number of data sets N, the mean absolute error MAE and the coefficient of determination R² are indicated in the individual panels.

L369-371: "As opposed to this, Fig. 6d shows that $RHi_{ANN}$ and $RHi_{IAGOS}$ have a closer agreement, with an MBE of about ±11% for all UT measurements up to 140%. The RHi between 80% and 130% in the important range for cirrus clouds is well represented by the ANN with an MBE better than +/- 7%."

[Figure]

Figure A4 (6): Frequency distribution (a and c) and overall mean biased error MBE (%) (b and d) of $RHi_{ERA5}$ and $RHi_{ANN}$ against $RHi_{IAGOS}$ in the clear and cloudy UT (grey) in the test data set.

[Figure]

Figure A6 (7): Distributions of IOT for contrails and cirrus retrieved from (a) MSG observations using the CiPS algorithm, and simulated using the CoCiP model with (b) $q_{ANN}$ or (c) $q_{IFS}$ at 10:00 UTC on 14 April 2021. The IOT distribution for contrails from CoCiP simulations is shown in (e) and (f). The linear structures of contrails correspond to the associated flight tracks. The IOT frequencies (histograms) for contrail cirrus and contrails are shown in (d) and (g), respectively.

[Figure]

Figure A7 (S6): RHi derived from (a) ERA5 or (c) the ANN model and the differences relative to AIMS measurements in (b) and (d) obtained from the HALO aircraft on 21 July 2021 during the CIRRUS-HL campaign. The lines present the HALO flight track.

References

Diao, M., Zondlo, M. A., Heymsfield, A. J., Avallone, L. M., Paige, M. E., Beaton, S. P., Campos, T., and Rogers, D. C.: Cloud-scale ice-supersaturated regions spatially correlate with high water vapor heterogeneities, Atmos. Chem. Phys., 14, 2639–2656, https://doi.org/10.5194/acp-14-2639-2014, 2014.

Diao, M., Jensen, J. B., Pan, L. L., Homeyer, C. R., Honomichl, S., Bresch, J. F., and Bansemer, A.: Distributions of ice supersaturation and ice crystals from airborne observations in relation to upper tropospheric dynamical boundaries, J. Geophys. Res.-Atmos., 120, 5101–5121, https://doi.org/10.1002/2015JD023139, 2015.

Dotzek, N. and Gierens, K.: Instantaneous fluctuations of temperature and moisture in the upper troposphere and tropopause region. Part 2: Structure functions and intermittency, Meteorol. Z., 17, 323–337, https://doi.org/10.1127/0941-2948/2008/0292, 2008.

He, K., Zhang, X., Ren, S., and Sun, J.: Delving Deep into Rectifiers: Surpassing Human-Level Performance on ImageNet Classification, 2015 IEEE International Conference on Computer Vision (ICCV), Santiago, Chile, 1026-1034, https://doi.org/10.1109/ICCV.2015.123, 2015.

**Responses to Community #2 Scott Geraedts**

We thank Scott Geraedts for the interest in the contingency table metrics used in our manuscript to evaluate ice supersaturation prediction and the helpful comments, which we address in the revision of the manuscript.

In the following we number the community's comments (CC) and our replies (R) to each, referencing the corresponding tracked changes in the manuscript.

**CC:** In addition to the ETS, it would be nice to have the full contingency table used to evaluate the model (e.g. for the cases in Table 2), so that other metrics could be computed if readers are interested.

**Ra)** author's response

Thank you for your interest in the complete contingency table metrics. The four events - whether ice supersaturation is predicted by ERA5 or the ANN model and observed by IAGOS – have already been explained in the discussion version of the manuscript. We have now added the relevant equations to the text, linking them to computed metrics based on the updated ANN model, which uses separate training, validation, and test data (see response letters to Anonymous Referee #2 and community #1, #3). Additionally, we have provided an updated table for the test data results. Please refer to Table A1 below, as well as Table S2 and the revised text in the manuscript.

**Rb)** manuscript changes

L396-400: "First, events are labeled according to the contingency table, with a ($Y_{IAGOS}/Y_{ERA5}$ or $Y_{IAGOS}/Y_{ANN}$, ice supersaturation predicted and observed), b ($Y_{IAGOS}/N_{ERA5}$ or $Y_{IAGOS}/N_{ANN}$, no ice supersaturation predicted but observed), c ($N_{IAGOS}/Y_{ERA5}$ or $N_{IAGOS}/Y_{ANN}$, ice supersaturation predicted but not observed), and d ($N_{IAGOS}/N_{ERA5}$ or $N_{IAGOS}/N_{ANN}$ ice supersaturation neither predicted nor observed). $Y_{IAGOS}$ indicates that the waypoint is in ice supersaturation based on the IAGOS measurements, while $N_{IAGOS}$ indicates the absence of ice supersaturation. The same notations are applied when analyzing the statistics for ERA5 and ANN."

L414-417: "The scores for ERA5 in all sky UT, cloudy UTLS, and all sky LS classes are 0.23, 0.21, and 0.14, respectively, indicating limited predictive skill, particularly in the all sky LS region. In contrast, the ANN model significantly enhances the ice supersaturation prediction, yielding scores of 0.71, 0.70, and 0.52 for the respective regions. This represents an approximate 0.44 increase in ETS across all classes…"

L428-429:" demonstrating a higher accuracy with an ETS as high as 0.71."

L491-492:" The dynamic-based humidity correction leads to an increase in ETS from 0.23, 0.21, and 0.14 (ERA5) to 0.71, 0.70, and 0.52 by the ANN, respectively..."

Table A1 (2): ETS values for the prediction of RHi>100% from $RHi_{ERA5}$ and $RHi_{ANN}$ in the test data set between 200 hPa and 400 hPa over the Atlantic, Europe and Africa in 2020.

| Scenarios | $Y_{IAGOS}/Y_{ERA5}$ | $Y_{IAGOS}/N_{ERA5}$ | $N_{IAGOS}/Y_{ERA5}$ | $N_{IAGOS}/N_{ERA5}$ | ETS |
|---|---|---|---|---|---|
| clear and cloudy UT | 66.34% | 3.31% | 19.42% | 10.94% | 0.23 |
| cloudy UTLS | 54.61% | 4.99% | 24.06% | 16.34% | 0.21 |
| clear and cloudy LS | 97.48% | 0.21% | 1.96% | 0.36% | 0.14 |
| clear sky UTLS | 95.43% | 0.05% | 4.23% | 0.28% | 0.06 |

| Scenarios | $Y_{IAGOS}/Y_{ANN}$ | $Y_{IAGOS}/N_{ANN}$ | $N_{IAGOS}/Y_{ANN}$ | $N_{IAGOS}/N_{ANN}$ | ETS |
|---|---|---|---|---|---|
| clear and cloudy UT | 67.07% | 2.57% | 4.57% | 25.79% | 0.71 |
| cloudy UTLS | 56.26% | 3.34% | 4.95% | 35.44% | 0.70 |
| clear and cloudy LS | 97.29% | 0.40% | 0.89% | 1.42% | 0.52 |
| clear sky UTLS | 94.99% | 0.49% | 2.07% | 2.44% | 0.47 |

**Responses to community #3 #4 Olivier Boucher**

We thank Olivier Boucher for the helpful comments on separating the training, validation, and test datasets; on input data considerations including meteorological variables, prior times and pressure levels; on the normalization of input data; and on setting variables to their mean values to assess changes in loss for evaluating relative importance in RHi prediction. We have addressed these points in the revised manuscript.

In the following we number the community's comments (CC) and our replies (R) to each, referencing the corresponding tracked changes in the manuscript.

**CC1:** The authors write that "This collocation of model meteorological variables and measured humidity values from the year 2020 comprises 3.99 million individual data points, from which 80%, 10%, and 10% are randomly selected for training, testing during the model development, and validating the ANNs, respectively." Presumably they refer to the full-resolution (i.e., 4s sampling) IAGOS data. Given the high sampling frequency, and if our understanding is correct, there is a strong autocorrelation in the data. Thus randomly selecting the training, testing and validation datasets implies that very similar conditions to those of the testing and validation datasets have been met in the training dataset. There is a well-known risk that this inflates artificially the model performance (see e.g. https://doi.org/10.1016/j.ophoto.2022.100018). At the very least the authors should select separate IAGOS flights in their training, testing and validation datasets. Even better they should consider dates that are at least one day apart for a given region.

It is unclear if the testing or validation dataset are used in Section 4 as text on line 185 and in Section 4 appears contradictory. In any case we would recommend the testing and validation datasets to be temporally disjoint from the training dataset at every location.

**R1a)** author's response

Thank you for commenting on the potential autocorrelation in IAGOS measurements and suggesting a further separation of the training, validation, and test dataset and ensure that the dates are at least one day apart for each given region.

We appreciate this feedback and calculated the autocorrelation function of the IAGOS data series following the method of Dotzek and Gierens (2008). The results, displayed in Figure A1 here, show that autocorrelation reaches up to 0.04 at the original resolution (4 s, 1 km) before gradually declining to near zero at a resolution of 0.25 degrees (25 km), which corresponds to the ERA5 grid box size. It can be explained by the nature that the water vapor field is quite chaotic with steep gradients. Diao et al (2014, 2015) showed that in-situ RHi measurements taken with 1 Hz or 10 Hz instruments reveal small-scale structures in the RHi time series, in agreement with our finding. Therefore, we assume that autocorrelation can be disregarded for the averaged IAGOS RHi measurements within the ERA5 grid box.

[Figure]

Figure A1: Autocorrelation function of the IAGOS measurement series (4s, 1km).

In response, we have still revised our methodology to ensure that the training set consists of data from time periods that do not overlap with those used for either validation and testing. Specifically, we are now using 4 days of data for building the ANN model, followed by a 1-day gap, and then 1 day of data for validation or testing. The complexity of the ANN model has also been increased to better match the growing complicacy of the training data, including the addition of hidden layers and neurons, as well as weight initialization and batch normalization for each layer to enhance generalization. Revisions to the text, along with updated Figures 4, 5, 6, 7, and S6 for RHi using the new models, are provided below. Updates on the validation of $T_{IAGOS}$ and $q$ predictions are presented in Figures S1, S3, and S4 in the revised supplement.

Thank you for pointing out the ambiguous description. The validation dataset is only used during the training process to tune model hyperparameters and check how well the model generalize to data it hasn't seen before. In the revised manuscript, we have changed the term in lines 193-194 from "validation dataset" to "test dataset."

**R1b)** manuscript changes

L174-176: "To ensure that the training set consists of data from time periods that do not overlap with those used for either validation and testing, we now use 4 days of data to build the ANN model, followed by a 1-day gap, and then 1 day of data for validation or testing."

L291-294: "In Sect. 2.3, four consecutive days of samples from the ERA5-IAGOS collection are allocated for model training, with the following day excluded to avoid overlap with the continuous weather system, and another day reserved either for validation, to evaluate the model's generalization to unseen data during training, or for testing."

L43-45 in the supplement: To ensure model robustness "and construct an independent test data set, we now use a sequence-based split: four consecutive days of data are used to build the ANN model, followed by a 1-day gap, with the subsequent day's data reserved for validation or testing".

L286-287: "We use 3 hidden layers, each with 100 neurons and He weight initializer (He et al., 2015), along with batch normalization between layers to improve generalization. The humidity output is referred to as $RHi_{ANN}$."

L298: "an ANN for $q$ is implemented, with 300 neurons in each hidden layer and…"

L18-20 in the Abstract: "The ANN shows excellent performance and the predicted RHi in the UT has a mean absolute error MAE of 5.7% and a coefficient of determination R² of 0.95, which is significantly improved compared to ERA5 RHi (MAE of 15.8%; R² of 0.66)."

L348-350: "The MAE decreases significantly from 15.82% (ERA5) to 5.71% (ANN), the $R^2$ values increase from 0.66 (ERA5) to 0.95 (ANN), and the root mean spare error RMSE decreases from 20.52% (ERA5) to 7.88% (ANN)."

[Figure]

Figure A2 (4): Distribution of (a) $RHi_{ERA5}$ and (b) $RHi_{ANN}$ versus $RHi_{IAGOS}$ in the UT in all sky (clear and cloudy conditions) in the test data set. The number of data sets N, the mean absolute error MAE and the coefficient of determination R² are shown in the panels.

L354-356: "In the cloudy (Fig. 5a-b) and clear sky (Fig. 5c-d) conditions in the UTLS, the MAE of the RHi decreases from 16.28% (11.21%) to 5.95% (4.28%), respectively. Also, the R² increases by 0.30 (0.23) to 0.95 (0.95) for the two scenarios."

L361-363: "The ANN model also has strong skills of RHi correction in the LS, see Fig. 5e and f. $R^2$ values increase from 0.59 (ERA5) to 0.95 (ANN), similar to the UT region in Fig. 4. The improvement of RHi prediction by the ANN is also documented by the decrease of MAE by 6.07%."

L377: "with a MAE of approximately 5.8%..."

L485-486: "Using this ANN humidity correction, the MAE of $RHi_{ANN}$ when comparing to $RHi_{ERA5}$ both against $RHi_{IAGOS}$ is reduced from 15.82% to 5.71%, 16.28% to 5.95%, 11.21% to 4.28%, and 9.78% to 3.71%..."

[Figure]

Figure A3 (5): Comparison of $RHi_{ERA5}$ (left column) and $RHi_{ANN}$ (right column) against $RHi_{IAGOS}$ in the (a) and (b) cloudy UTLS, (c) and (d) clear sky UTLS, and (e) and (f) clear sky and cloudy (or all sky) LS regions in the test data set. The number of data sets N, the mean absolute error MAE and the coefficient of determination R² are indicated in the individual panels.

L369-371: "As opposed to this, Fig. 6d shows that $RHi_{ANN}$ and $RHi_{IAGOS}$ have a closer agreement, with an MBE of about ±11% for all UT measurements up to 140%. The RHi between 80% and 130% in the important range for cirrus clouds is well represented by the ANN with an MBE better than +/- 7%."

[Figure]

Figure A4 (6): Frequency distribution (a and c) and overall mean biased error MBE (%) (b and d) of $RHi_{ERA5}$ and $RHi_{ANN}$ against $RHi_{IAGOS}$ in the clear and cloudy UT (grey) in the test data set.

[Figure]

Figure A6 (7): Distributions of IOT for contrails and cirrus retrieved from (a) MSG observations using the CiPS algorithm, and simulated using the CoCiP model with (b) $q_{ANN}$ or (c) $q_{IFS}$ at 10:00 UTC on 14 April 2021. The IOT distribution for contrails from CoCiP simulations is shown in (e) and (f). The linear structures of contrails correspond to the associated flight tracks. The IOT frequencies (histograms) for contrail cirrus and contrails are shown in (d) and (g), respectively.

[Figure]

Figure A7 (S6): RHi derived from (a) ERA5 or (c) the ANN model and the differences relative to AIMS measurements in (b) and (d) obtained from the HALO aircraft on 21 July 2021 during the CIRRUS-HL campaign. The lines present the HALO flight track.

References

Diao, M., Zondlo, M. A., Heymsfield, A. J., Avallone, L. M., Paige, M. E., Beaton, S. P., Campos, T., and Rogers, D. C.: Cloud-scale ice-supersaturated regions spatially correlate with high water vapor heterogeneities, Atmos. Chem. Phys., 14, 2639–2656, https://doi.org/10.5194/acp-14-2639-2014, 2014.

Diao, M., Jensen, J. B., Pan, L. L., Homeyer, C. R., Honomichl, S., Bresch, J. F., and Bansemer, A.: Distributions of ice supersaturation and ice crystals from airborne observations in relation to upper tropospheric dynamical boundaries, J. Geophys. Res.-Atmos., 120, 5101–5121, https://doi.org/10.1002/2015JD023139, 2015.

Dotzek, N. and Gierens, K.: Instantaneous fluctuations of temperature and moisture in the upper troposphere and tropopause region. Part 2: Structure functions and intermittency, Meteorol. Z., 17, 323–337, https://doi.org/10.1127/0941-2948/2008/0292, 2008.

Kattenborn, T., Schiefer, F., Frey, J., Feilhauer, H., Mahecha, M. D., and Dormann, C. F.: Spatially autocorrelated training and validation samples inflate performance assessment of convolutional neural networks, ISPRS Open Journal of Photogrammetry and Remote Sensing, 5, 2667–3932, https://doi.org/10.1016/j.ophoto.2022.100018, 2022.

He, K., Zhang, X., Ren, S., and Sun, J.: Delving Deep into Rectifiers: Surpassing Human-Level Performance on ImageNet Classification, 2015 IEEE International Conference on Computer Vision (ICCV), Santiago, Chile, 1026-1034, https://doi.org/10.1109/ICCV.2015.123, 2015.

**CC2:** Line 268: it is not clear what the 56 inputs consist of. 8 times 2 times 5 makes 80 so presumably some variables have fewer times or pressure levels. Which ones? Table 1 does not really clarify this. Could the authors provide more details?

**R2a)** author's response

Thank you for marking this unclear sentence. The 56 inputs consist of 8 meteorological variables across 7 conditions: the current time and level, two time lags (-2h and -6h), and four ERA5 pressure levels surrounding the IAGOS cruise level (-2, -1, +1, +2). Additionally, we have revised the text and clarified the first explanatory column in Table A1(1) for better detail.

**R2b)** manuscript changes

L280-283: "The ANN model consists of 56 inputs, derived from 8 meteorological variables across 7 conditions: the current time and level, two time lags (-2 h and -6 h) for the current level and four ERA5 pressure levels surrounding the IAGOS cruise altitude (-2, -1, +1, +2 levels) for the current time."

Table A1(1): Overview of the variables used in this study. Spatial resolution of ERA5: 0.25°. Vertical resolution of ERA5 on pressure levels: 25-50 hPa. The original temporal resolution of ERA5 and IAGOS: 1 h and 4 s. Study regions: Atlantic, Europe and Africa.

| source | variable (* RHi ANN) | description | unit |
|---|---|---|---|
| ERA5 (7 conditions)
• the current time and level
• two time lags (-2h and -6h)
• four ERA5 pressure levels surrounding the IAGOS cruise level (-2, -1, +1, +2) | $* T_{ERA5}$ | air temperature | K |
| | $ciwc$ | specific cloud ice water content | Kg/kg |
| | $* RHi_{ERA5}$ | relative humidity w.r.t. ice | % |
| | $q_{ERA5}$ | specific humidity | g/Kg |
| | $* z$ | geopotential | $m^2/s^2$ |
| | $* w$ | vertical velocity | Pa/s |
| | $* d$ | divergency of wind | $s^{-1}$ |
| | $* u$ | eastward component of wind | m/s |
| | $* v$ | northward component of wind | m/s |
| | $* vo$ | relative vorticity | $s^{-1}$ |
| | $pv$ | potential vorticity | $s^{-1}$ |
| | time | hour | 1 |
| | level | pressure | hPa |
| IAGOS
at the current time | $* RHi_{IAGOS}$ | relative humidity w.r.t. ice | 1 |
| | $T_{IAGOS}$ | air temperature | K |
| | pressure | air pressure | Pa |

**CC3:** ANN: were the input and output data normalized and how? I could not find the information neither in the main text nor in the Appendix (sorry if I missed it). Section 3.3 presents an ablation study where one ERA5 variable is set to zero at a time, so I assume the data have been centred (as usual practice is to set the variable to its mean value).

**R3a)** author's response

Thank you for your questions regarding this completed normalization step but previously overlooked description. Yes, we apply min-max normalization to both input and output data in the deep learning pre-processing stage. This data pre-processing approach rescales data features to a fixed range of [0, 1], preserving the relationships between data points. This approach not only improves model performance by preventing features with larger ranges from dominating but also enhances convergence speed in gradient-based algorithms, such as the neural networks used in our method. Certainly, the model prediction results were transformed back to their original scale for testing by applying the inverse of the normalization, using the previously saved scaler. We have added this explanation to the main text.

Regarding the relative contributions of individual variables to the ANN's RHi predictions, we agree with your suggestion to set each variable to its mean value to assess its significance to RHi. Please refer to the updated Figure 3 in the manuscript or Figure A7 below. We have also revised the description of the changes in $K_x$ in the updated text.

**R3b)** manuscript changes

L285-286: "We apply min-max normalization to both input and output data, which prevents features with larger ranges from dominating and improves convergence speed during model training."

L295-296: "To test the model's predictions, the results were transformed back to their original scale by applying the inverse of the normalization using the previously saved scaler."

L306-307: "$K_x$ is the relative change in loss when one input, i.e. one feature of ERA5 is set to its mean value for the complete input data set" …

L309-310: "where $L_x$ is the loss (MSE) for the test data set compared with IAGOS when setting one ERA feature input to its average value" …

L18 in the Abstract: "while other dynamical variables are of low to moderate or high importance."

L321-323: "they provide a moderate and non-negligible contribution to the accuracy of the RHi prediction model. In fact, $w$ and $d$ show $K_x$ of 0.29 and 0.26 and those for $u$, $v$, and $vo$ even higher, which are 0.98, 0.75, and 0.74. There is generally less importance in the contributions of the variables representing dynamical quantities…"

L325-326: "The fact that dynamical variables for instance particularly u, v, vo are closely as important as $RHi_{ERA5}$, $T_{ERA5}$, and $z$ for the description of the physical processes that lead to the decrease/increase of relative humidity in Sect. S2 in the supplement…"

L329-332: "Hofer et al. (2024) shows that $RHi_{ERA5}$ is the most influential predictor for humidity predictions, while the explanatory power of dynamical proxies is insufficient when only using data from the current time and level. However, our updated analysis confirms that incorporating a broader vertical region and the historical time into the dynamical variables has a more significant impact on the ANN model and contributes to the understanding of humidity evolution."

L482-483: "while other meteorological variables, including horizontal wind speed, relative vorticity, vertical velocity, and divergence, have a high or moderate to minor but measurable influence."

[Figure]

Figure A7 (3): Relative importance of the individual variables to the ANN model for predicting RHi.